# Detection and Consequences of Atmospheric Deserts: Insights from a Case Study

Fiona Fix[1], Georg Mayr[1], Achim Zeileis[2], Isabell Stucke[1], and Reto Stauffer[3]

[1]Department of Atmospheric and Cryospheric Sciences, Universität Innsbruck, Innsbruck, Austria
[2]Department of Statistics, Universität Innsbruck, Innsbruck, Austria
[3]Department of Statistics & Digital Science Center, Universität Innsbruck, Innsbruck, Austria

**Correspondence:** Fiona Fix (fiona.fix@uibk.ac.at)

**Abstract.** We introduce the concept of atmospheric deserts (ADs), air masses that are advected away from hot and dry convective boundary layers in semi-arid or desert source regions. They can be expected to eliminate cloudiness, cause heat to build up in the target region, suppress thunderstorm formation in their centre and boost thunderstorm formation at their edges. A direct detection method, tracing the AD from source to target using Lagrangian trajectories is developed.

We illustrate this new concept of ADs and the application of the detection method with a case study in Europe from mid-June 2022. With the Lagrangian analysis tool (LAGRANTO) approximately 45 million trajectories are calculated, tracking the path of the air mass and the development of its properties as it progresses from North Africa towards and across Europe over the course of five days. $k$-means-clustering identifies four typical pathways that the trajectories follow. For one of the pathways, the air nearly conserves its well-mixed properties. Diabatic processes of radiative cooling, latent heating due to condensation, and cooling due to re-evaporation of precipitation, however, modify the air along the other pathways.

In this case in June 2022, thunderstorms were mainly absent in the centre of the AD, but broke out along a line parallel to its boundary. At this edge of the AD and the surface front, lifting occurred, causing the formation of thunderstorms. The AD did not reside directly above the local boundary layer for long enough to be the main cause for the high near-surface temperatures in large parts of Europe, but may have contributed to it. Subsidence heating of another air stream, was identified as one possible reason for the increased near-surface temperatures. This case supports the assumption that ADs co-occur with thunderstorms at their edges and increased near-surface temperatures in their centre and gives some insights into the responsible processes.

## 1 Introduction

Severe thunderstorms and heat waves bear serious risks for human health, economy, and society. Heat waves are the reason for many deaths (e.g., Schär and Jendritzky, 2004; Schär, 2016) especially in highly populated regions like Europe, and thunderstorms can cause severe economic or ecologic damage. Understanding weather situations that influence these extreme weather events is therefore of great importance.

This paper introduces ADs as air masses that originate in the hot and dry convective boundary layers (CBLs) of semi-arid, desert, subtropical and/or elevated source regions. They can strongly influence the vertical temperature and moisture profiles in

the regions they are advected into and create large temperature and moisture gradients at their lateral boundaries. We illustrate a direct detection method of ADs in this study and present first insights gained from a case study from mid-June 2022.

Previous research has studied a small subset of possible manifestations of ADs by looking for well-mixed, warm, and dry layers on top of the local boundary layer (BL) in vertical profiles in the target region (e.g. Carlson and Ludlam, 1968; Carlson et al., 1983; Lanicci and Warner, 1991a; Banacos and Ekster, 2010; Cordeira et al., 2017; Ribeiro and Bosart, 2018), sometimes complemented by numerical weather prediction models (Arritt et al., 1992) or satellite imagery (Gitro et al., 2019). These layers are termed "elevated mixed layers" (EMLs, e.g. Carlson et al., 1983; Banacos and Ekster, 2010; Ribeiro and Bosart, 2018, and others). They occur in the special case where the thermodynamic properties of the AD remain (almost) constant during the advection. More commonly, however, diabatic processes will modify the ADs along their way. These modifications together with differential advection in the vertical will make the air mass unrecognizable in the vertical profiles in the target region. We therefore expect EMLs to be (much) rarer than their generalization, the ADs.

EMLs are known to greatly impact heat wave and thunderstorm formation (e.g. Carlson and Ludlam, 1968; Carlson, 1980; Carlson et al., 1983; Farrell and Carlson, 1989; Banacos and Ekster, 2010; Cordeira et al., 2017; Dahl and Fischer, 2016; Lanicci and Warner, 1991a, b; Ribeiro and Bosart, 2018), and we conjecture that ADs can have similar consequences (evidence was found by Johns and Dorr, 1996), although possibly of smaller magnitude. These consequences stem from strong temperature and moisture gradients at the vertical and lateral boundaries of ADs (e.g. Carlson et al., 1983; Farrell and Carlson, 1989; Dahl and Fischer, 2016). For the special case of EMLs it was found that the hot and dry air masses ride up over the cooler, moister, shallower CBL in the target region, and can form a capping inversion (or "lid", e.g., Carlson and Ludlam, 1968; Carlson, 1980; Carlson et al., 1983; Lanicci and Warner, 1991a, b; Cordeira et al., 2017) and contribute to potential instability. The lid can lead to heat buildup underneath, especially under the typically associated cloud free conditions, which leads to an increase in the convective available potential temperature (CAPE) and the near-surface temperatures (e.g., Carlson and Ludlam, 1968; Carlson, 1980; Carlson et al., 1983; Keyser and Carlson, 1984; Farrell and Carlson, 1989; Cordeira et al., 2017). It was found that (severe) thunderstorms are often initiated along the edge of the EML (Carlson and Ludlam, 1968; Carlson et al., 1980, 1983; Keyser and Carlson, 1984; Lanicci and Warner, 1991c; Arritt et al., 1992; Banacos and Ekster, 2010; Lewis and Gray, 2010; Sibley, 2012; Dahl and Fischer, 2016; Cordeira et al., 2017, and others). Towards the edge of the EML, the lid base height increases and its strength decreases, hence the constraint on convection at the edge is not as strong as in the central area, where the lid suppresses thunderstorm initiation (Carlson et al., 1983). In some cases ADs may also bring dust from the source to the target region, however, this is not the focus of this study.

The consequences of EMLs and ADs can be expected to be similar, however, the latter has never been studied before. Hence this study is looking at one case of an AD that would not have been classified as an EML, but that co-occurred with strong lightning activity along its edge and high near-surface temperatures in its centre.

Since more than one third of the Earth's land surface is arid (e.g. Vaughn, 2005; Tchakerian, 2015; European Commission et al., 2018), ADs might play an important, yet understudied role in mid-latitude weather. This study introduces the new concept of ADs and a universal direct way to identify them and trace their properties along their path. This is described in general terms in Section 2. The application of the detection method and the impact of an AD on the weather in the target region are illustrated

using a case study of an AD that originates in North Africa and travels across Europe during a five-day period in June 2022. The data and trajectory model used for this case study are described in Section 3.1, before the case study and the details on the detection are given in Section 3.2. The results on the modification of the air mass enroute and the consequences for the weather in the target region are presented in Sections 3.3 and 3.4, respectively.

## 2   Definition of atmospheric deserts and detection method

ADs are air masses that are advected away from the hot and dry CBL of semi-arid, desert, subtropical and/or elevated source regions. These air masses progressively lose their distinct characteristics during the advection over hundreds to thousands of kilometers due to diabatic processes and differential advection in the vertical. A special case of an AD is an EML, in which case the air mass remains (almost) unmodified and well-mixed.

As ADs are generally modified during the advection, indirect detection methods based on the properties in the target region are ambivalent and often insufficient. Therefore, we introduce a novel detection method, that traces the air mass directly from its source to the target region, using Lagrangian trajectories. The AD is then defined as all the grid boxes (any grid chosen for the application at hand) that contain at least one trajectory at a given time. The development of the AD along its path can be analyzed. While this may seem to be a weak definition, it is a useful one. The number of trajectories that reach a cell is highly dependent on the number of initiated trajectories, which may differ depending on the application and the available computing resources. An analysis of the case study at hand shows that typically the number of trajectories per cell is much higher than one, so that we do not expect to substantially misidentify the AD cells and especially edges by using this criterion.

The detection method requires a trajectory calculation tool and a spatio-temporally complete data set of atmospheric data over a large area covering the source and the target regions. A high vertical resolution of the meteorological data is crucial for the detection and analysis of the ADs.

### 2.1   Trajectory calculation

The AD air is traced directly from the source to the target region. Hence, forward trajectories need to be initiated in the source region. Per definition, the origin of an AD is the CBL of an arid source region. In these regions, the depth of the CBL can reach up to several kilometres during daytime, especially in the summer months (Garcia-Carreras et al., 2015). The nocturnal boundary layer in desert regions can become very shallow with deep overlying residual layers from the daytime CBL. Since ERA5 data only contain the height of the (stable) nocturnal boundary layer but not of the residual layer, we ran sensitivity experiments with different methods of deriving the top of the residual layer from ERA5. We found that only using trajectories started during the hottest hours of the day between 13:00 and 17:00 UTC (inclusive) yielded similar results as with additional nocturnal trajectories started in the residual layer. Meteorological variables are traced along the trajectories. The spatio-temporal resolution of the initialization, the length of the trajectories, and the size of the grid boxes depend on the application and the available data. For the application in this case study, details are given in Section 3.1.

## 2.2 Trajectory clustering

In order to simplify the analysis of very many trajectories, we group them into several clusters, representing *typical trajectory pathways*. Identifying typical pathways requires defining the characteristics for comparison. Here, we consider spatial (longitude, latitude, altitude), thermodynamic, and microphysical aspects. We employ the 11 variables listed in Tab. 1 and the data driven $k$-means-clustering method (MacQueen, 1967) to cluster the trajectories. This is a multivariate clustering approach, using the 11-dimensional data after normalizing all variables (zero mean and unit variance) in order to give all of them the same "weight" or "importance" in the clustering. A similar approach was also used by Nie and Sun (2022). This method identifies $k$ clusters (groups) in the data, where the data points within one cluster are as similar as possible, while the clusters are as dissimilar as possible. The measure for the similarity is the squared Euclidean distance between each data point and the cluster means.

To capture the spatial aspects, the differences between the final and starting positions of the trajectories are calculated for longitude, latitude, and height above mean sea level ($hamsl$): $\text{diff}_{\text{lon}}$, $\text{diff}_{\text{lat}}$, and $\text{diff}_{hamsl}$. Additionally, the maximum difference in $hamsl$ in all 6 h windows along the trajectory ($\text{diff}_{hamsl,\max}$) is useful to distinguish between trajectories that rise slowly and those that rise abruptly. To account for subsidence after an initial ascent, we introduce the difference between the maximum and the final height, $\text{diff}_{hamsl,\text{subs}}$. Since different diabatic processes such as radiation, mixing, or condensation/evaporation have different impacts on the changes of the thermodynamic and microphysical variables, the differences of the potential temperature $\theta$ ($\text{diff}_{\theta}$), the specific humidity $q$ ($\text{diff}_q$) and the total (logarithmic) water content $cwc$ ($\text{diff}_{cwc}$) between the final and starting locations are taken into account. Here, $cwc$ is the sum of the four types of cloud water content: liquid ($clwc$), ice ($ciwc$), rain ($crwc$), and snow ($cswc$). Cloud water content variable distributions are typically heavily skewed towards low values. This skewness can be reduced by performing a logarithmic transformation. The diabatic processes do not only affect the differences in the thermodynamic and microphysical variables, but also on their evolution. Some processes lead to correlated changes in temperature and moisture, for example, while for other processes the changes are unrelated. Hence, the correlation between the potential temperature and the specific humidity ($\text{corr}_{\theta,q}$) and the correlations between the precipitating and non-precipitating cloud water contents and the 6-hourly differences in the potential temperature are taken into account. Here, we refer to the precipitating cloud water content ($cpwc$) as the sum of $crwc$ and $cswc$, and the non-precipitation cloud water content ($ccwc$) as the sum of $clwc$ and $ciwc$. The respective correlations are denoted as $\text{corr}_{\text{diff}_{\theta},cpwc}$ and $\text{corr}_{\text{diff}_{\theta},ccwc}$. All these variables are summarized in Tab. 1 and need to be normalized by subtracting their mean and dividing by their standard deviation before applying the clustering method, so that those with large values do not dominate the clustering.

The cluster averages can then be used for analyzing the development of the properties along the different *typical pathways*. The mean is calculated in the trajectory relative time frame (hours since initialization), rather than in the absolute time frame, since trajectories typically move faster than the synoptic situation, so that this yields a more comparable result. Note that this average is not a trajectory itself and might not be physically consistent across variables. An exemplary figure and a more detailed description of the effects of averaging can be found in Appendix A (Fig. A1).

| Variable | Formula |
|---|---|
| $\text{diff}_{\text{lon}}$ | lon at the final location – lon at the starting point |
| $\text{diff}_{\text{lat}}$ | lat at the final location – lat at the starting point |
| $\text{diff}_{hamsl}$ | $hamsl$ at the final location – $hamsl$ at the starting point |
| $\text{diff}_{hamsl,\text{max}}$ | maximum(6-hourly differences in $hamsl$) |
| $\text{diff}_{hamsl,\text{subs}}$ | maximum($hamsl$) – $hamsl$ at final location |
| $\text{diff}_{\theta}$ | $\theta$ at the final location – $\theta$ at the starting point |
| $\text{diff}_{q}$ | $q$ at the final location – $q$ at the starting point |
| $\text{diff}_{cwc}$ | $(\ln(ciwc) + \ln(cswc) + \ln(clwc) + \ln(crwc))$ at the final location |
| | $- (\ln(ciwc) + \ln(cswc) + \ln(clwc) + \ln(crwc))$ at the starting point |
| $\text{corr}_{\theta,q}$ | correlation($\theta$, $q$) |
| $\text{corr}_{\text{diff}_{\theta},ccwc}$ | correlation(6-hourly differences of $\theta$, $\ln(ciwc)$) + correlation(6-hourly differences of $\theta$, $\ln(clwc)$) |
| $\text{corr}_{\text{diff}_{\theta},cpwc}$ | correlation(6-hourly differences of $\theta$, $\ln(cswc)$) + correlation(6-hourly differences of $\theta$, $\ln(crwc)$) |

**Table 1.** Variables characterizing the trajectories, used in $k$-means-clustering. For further explanation regarding their meaning, refer to the text.

## 3 Practical Application

### 3.1 Data and trajectory model

The model chosen to calculate the trajectories in this work is the Lagrangian analysis tool (LAGRANTO) version 2.0, which has been developed since the late nineties (Sprenger and Wernli, 2015), and is therefore a mature and widely used tool in the atmospheric sciences, in various contexts (e.g. Stohl et al., 2001; van der Does et al., 2018; Keune et al., 2022; Oertel et al., 2023). Forward or backward trajectories can be calculated iteratively based on the 3D wind field of the input dataset with customized starting region and resolution. In this study, Europe is the target region, hence North Africa is used as the source

region. A polygon marking the source region can be seen as the grey outline in Fig. 1. This source region lies completely in an arid, desert, hot climate zone (Beck et al., 2018), avoids coastal regions, and is in the lee of the Atlas mountains for the flow patterns causing ADs. Earlier studies took the Iberian peninsula as the source region for European EMLs (e.g. Carlson and Ludlam, 1968; Karyampudi and Carlson, 1988; Lewis and Gray, 2010; Dahl and Fischer, 2016), however we find that including Iberia only increases the number of trajectories by a few percent and the influences on the results are marginal. Trajectories are

started at a very high resolution of 5 km in the horizontal and 10 hPa in the vertical between 1100 and 400 hPa, from below the BLH between 13:00 and 17:00 UTC. They are calculated 120 h forward in time.

As a spatio-temporally complete data set of atmospheric data, we use the latest global reanalysis from the European Centre for Medium-Range Weather Forecasts (ECMWF), ERA5 (Hersbach et al., 2020), which is based on the Integrated Forecasting System (IFS) Cy41r2. The horizontal resolution of ERA5 is 0.25°, and data are available hourly on 137 vertical model levels up

to 1 Pa (Hersbach et al., 2020). This results in a high vertical resolution of about 20 m at the surface and 300 m at 500 hPa. The

domain chosen for this study covers Northern Africa and Europe, specifically 30° W to 60° E and 15° to 73° N. ERA5 single-level, pressure-level and model-level data on the lowest 74 model levels are obtained (surface to about 120 hPa). Additional to the reanalysis variables, we obtain the mean temperature tendency due to short and long wave radiation ($mttswr, mttlwr$), which are only available as forecast variables. We use the forecast from 06:00 UTC for the times from 09:00 UTC to 20:00 UTC and the forecast from 18:00 UTC for the times between 21:00 UTC and 08:00 UTC to avoid the respective spin-up periods. Aerosol optical depth at 550 nm is acquired to estimate whether the AD transports dust (0 h leadtime for the 00:00 and 12:00 UTC forecast, CAMS; European Centre for Medium-Range Weather Forecasts, 2023).

According to the IFS documentation (European Centre for Medium-Range Weather Forecasts, 2016), the different types of water content can be converted into one another by condensation, melting, autoconversion, etc. The rain, snow, and ice particles ($crwc, cswc, ciwc$), are allowed to sediment. Constant fall speeds of 4, 1, and 0.13 ms$^{-1}$ are assumed, respectively. Precipitation is allowed to be advected by the 3D-wind, and to re-evaporate when falling through an environment with lower relative humidity than a critical value. For more detail the reader is referred to the model documentation (European Centre for Medium-Range Weather Forecasts, 2016; Hersbach et al., 2020).

The trajectories are aggregated to grid boxes of 0.25° times 0.25° times 500 m, matching ERA5 grid cells in the horizontal. An AD-cell is a grid box that contains at least one trajectory. Lightning is used as a proxy for thunderstorm location, hence a lightning measurement data set is suitable to analyse the connection between ADs and convection. This study uses data from the lightning location network "Blitzortung" (Wanke et al., 2014). The network processes data from sensors operating at the very low frequency range set up by a large number of volunteers around the world. In this frequency range, weaker strokes are not detected and detection efficiency varies slightly between day and night. However, the network still allows to reliably detect locations of more widespread lightning activity. Consequently, those records where only a single flash was observed within a radius of one ERA5 grid cell were omitted.

## 3.2 Exemplary case study

A case study in June 2022 is used as an example to explain the concept of ADs and first findings in this study. Central Europe experienced intense heat during the period between 18 June 2022 and 19 June 2022 (e.g. Imbery et al., 2022, also see Fig. 3). Intense heat was recorded in southern Europe already in the days prior. Low pressure systems were located so that advection of air from Northern Africa to Europe occurred.

The five-day period from 15 to 19 June 2022 is chosen for this case study. Trajectories are initiated during daytime hours (13:00 through 17:00 UTC) hourly between 15 June 2022 00:00 UTC and 19 June 2022 11:00 UTC. This results in approximately 45 million trajectories starting from the North African BL during this case study. About 80% of the trajectories never pass north of 37° N. About 1.5% have left the domain by 12 UTC on 19 June 2022. The remaining 8.7 million pass north of 37° N at least once and have not left the domain yet by 19 June 2022 12 UTC and are therefore interesting for further analysis.

As the synoptic situation changes during the period of the case study, trajectories initiated at different times will follow different pathways. However, we do not expect big differences during one day, hence we chose to apply the clustering on all trajectories that start on the same day. We standardize the variables and apply the clustering method explained in Sect. 2.2

on each initialization day separately. 4.79 million/3.1 million/800 thousand/285 trajectories are clustered for the initialization days 15/16/17/18 June 2022, respectively. In this case, four clusters are chosen, as the total sum of squares does not decrease drastically for more clusters. Note that due to the different lengths of the trajectories, during the last 23 h since initialization, fewer trajectories are averaged, when calculating the cluster average. Of all the grid boxes north of $37°$ N that were identified as AD cells at 19 June 2022 12 UTC, the respective clusters C1 to C4 (of all days combined) cover 36.55 %, 58.85 %, 22.08 %,

and 14.96 %, respectively.

Figure 1 shows the evolution of the geopotential, AD maximal extent, AD extent at 800 hPa, and 800 hPa fronts during the period 16–19 June 2022. A low pressure system, initially located south west of the Iberian peninsula, moves north east, across the Gulf of Biscay during the case study. Air from North Africa is advected northwards in the northeasterly current east of this low pressure system. Another low pressure system travels from south of Greenland to southwestern Scandinavia, its cold front is shown in blue in Fig. 1. About 40 % of all ERA5 grid cells north of $37°$ N (within $30°$ W to $60°$ E and $37°$ to $73°$ N) are

185 covered by the AD for at least for one hour during the study period. By the end of the studied period, large parts of Europe are covered by the AD (see Fig. 1, at June 19 2022 12 UTC, 28 % of all columns within the domain $30°$ W to $60°$ E and $37°$ to $73°$ N are covered by the AD), which extends as far north as the British Isles and southern Sweden and as far east as Russia, north of the Black Sea. AD-cells can be found at any altitude between the ground and up to 13 km. The majority of identified

AD-cells at 12 UTC on 19 June 2022 is at 3.5 km. The black outline of the maximum extent of the AD in Fig. 1 shows that while the entire AD still resides south of the 800 hPa cold front at 18 June 2022 00:00 UTC, its edge passes north of it during the next day. However, the edge of the AD at the 800 hPa level remains south of the cold front for longer, and the front only catches up with it in parts by the end of the case study. The lateral boundary of the AD is also marked by a strong horizontal temperature gradient that sharpens over time but remains separate from synoptic frontal boundaries.

Yellow crosses in Fig. 1 mark locations where lightning occurred in the 2 h window centred at the given time. Thunderstorms persist in the vicinity of the occluding front west of the Iberian peninsula and in the Gulf of Biscay. Lightning also occurs along the edge of the AD. Especially during the night of 19 June 2022, there is a distinctive line of lightning close to the edge of the 800 hPa AD-layer. Notably, most of the area covered by the AD, especially in its centre, does not experience any thunderstorm activity. This is analyzed more closely in Sect. 3.4.3. Additionally, in this case the dust aerosol optical depth (550 nm) is notably

increased in the area covered by the AD, and especially elevated in its centre. This indicates that the AD brings Sahara dust to the target region, which is, however, not further discussed in this paper.

## 3.3 Modification of the thermodynamic properties

The cluster averages for the 4.79 million trajectories started on 15 June 2022 are shown in Fig. 2. Averages are calculated over 1.6, 1.8, 0.89, and 0.56 million trajectories for clusters C1, C2, C3, and C4, respectively. The figure shows the development of

205 different thermodynamic variables with time since initialization (panels a–f), as well as the path across the map (Fig. 2g).

During the first 24 to 36 h, all four clusters behave similarly and rise from approximately 2 km to approximately 4 km (Fig. 2a), while their potential temperature increases (Fig. 2b), and the absolute temperature decreases (Fig. 2c). During this time the trajectories still reside in North Africa (Fig. 2g). The cumulative mean temperature tendency due to radiation (Fig. 2b,

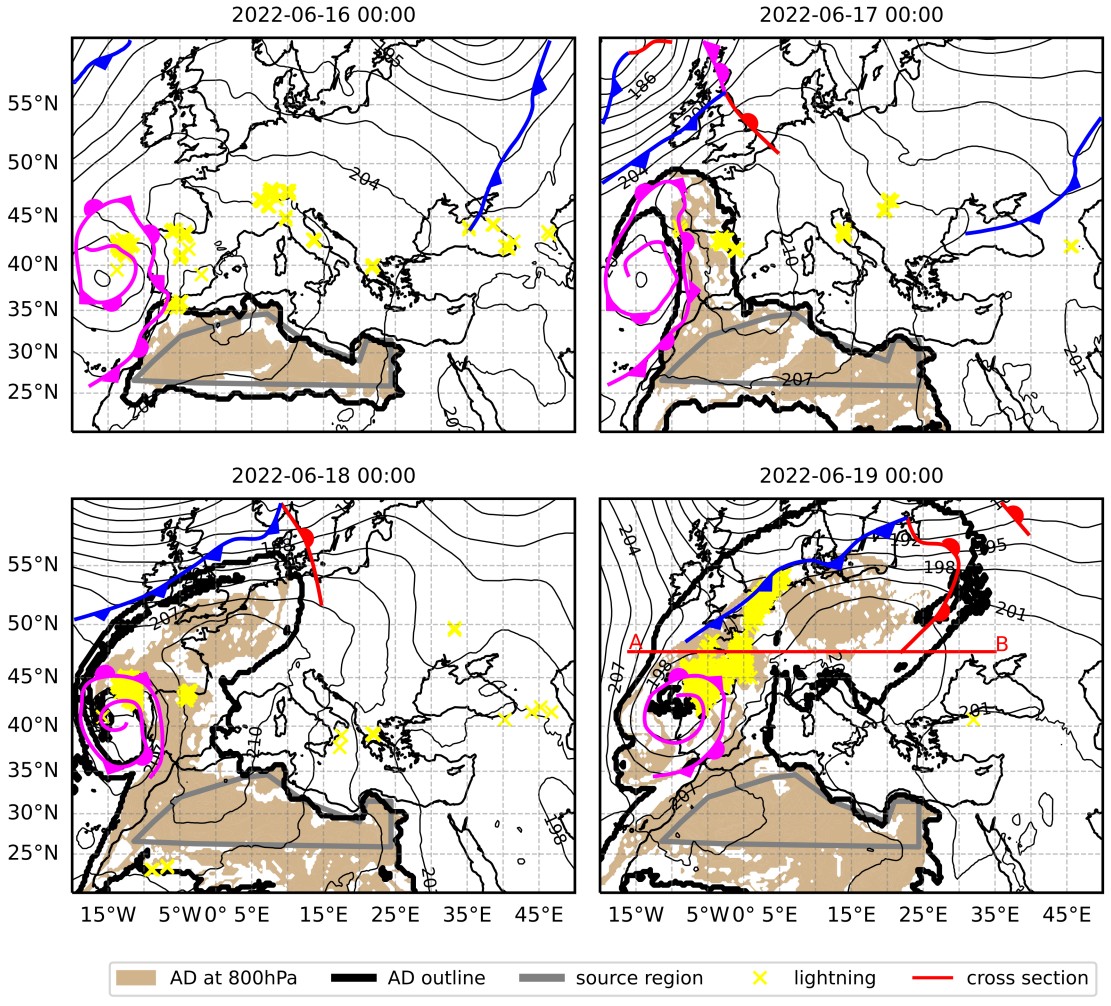

**Figure 1.** Display of the situation during 16–19 June 2022, 00:00 UTC, respectively. Thin black contours show the 800 hPa geopotential height in decametres, with a spacing of 3 dam. The coloured lines denote the 800 hPa fronts (identified from 800 hPa temperature, relative humidity, divergence, and relative vorticity maps), colours and symbols have their usual meaning. The maximum extent of the AD is outlined in thick black. The extent of the AD in the layer from 800 to 750 hPa is marked in beige. Yellow crosses mark locations where lightning occurred during the hour before and after. Red line (A–B) marks the location of the cross-section depicted in Fig. 4.

dashed) does not explain this diabatic warming, and the specific humidity does not indicate that latent heating is responsible, hence it can be assumed that mixing with a warmer air mass, such as the local CBL, is responsible for this increase in potential temperature.

After these initial 24 to 36 h the clusters begin to differ: Clusters C1, C2, and C3 (cyan, blue, and red) follow a very similar geographical path, crossing the Iberian peninsula, turning east over the Gulf of Biscay and travelling further east across Northern France and Germany towards Eastern Europe (Fig. 2g). Cluster C1 follows what could be called the typical or expected behaviour of an EML. It rises slightly (Fig. 2a) while riding up on local air masses, but it almost conserves its potential temperature (Fig. 2b) and specific water vapour content (Fig. 2d), which would preserve the well-mixed properties of the North African CBL.

Diabatic processes, however, modify the properties of the trajectories in the other clusters. The trajectories in Cluster C2 (blue) rise much higher on average (up to approximately 8 km, Fig. 2a). An especially sudden ascent is visible around hour 80. During this ascent, the trajectories cool adiabatically (Fig. 2c), which induces condensation (decrease in $q$, Fig. 2d, and increase in cloud water content variables, Fig. 2e,f). Latent heat causes the potential temperature in this cluster to rise (Fig. 2b) and condensate precipitates out (the total water content $q_t$ decreases, not shown here, as it is almost indistinguishable from $q$ in Fig. 2d). This sudden ascent coincides with the location where the cluster rises on top of the cooler air mass in the north (Figs. 1 and 2g). With this behaviour, this cluster is the most similar to a warm conveyor belt (e.g. Browning, 1971).

In contrast, cluster C3 (red) remains at a constant height above mean sea level after the ascent during the initial 24 h and then experiences a descent around hour 80. Meanwhile, its potential temperature decreases (Fig. 2b) and its specific water vapour content increases (Fig. 2d). This is partly due to radiative cooling (dashed in Fig. 2b), and partly due to evaporative cooling as precipitation falling through from above re-evaporates. This explanation is supported by the fact that together with the decrease in potential temperature (Fig. 2b), the specific water vapour content increases (Fig. 2d). Re-evaporation is possible in the data used here, since ice, snow and rain are allowed to sediment and can re-evaporate when they fall through a sub-saturated air mass in ERA5 (European Centre for Medium-Range Weather Forecasts, 2016). The strong correlation between the precipitation cloud water contents of C2 and C3 (dashed in Fig. 2e, f) together with the increase of specific water content (Fig. 2d) indicates it may be the precipitation from C2 that re-evaporates in C3. Also, mixing with the cooler, moister local air can be a reason for the cooling and moistening. As the cluster average is comprised of many different trajectories, it is likely that all three processes play a role.

The trajectories in the fourth cluster (C4, orange) take longer to leave North Africa and turn east already over the Iberian peninsula on average, so that they almost reach the Mediterranean Sea. A closer analysis of the trajectories in this cluster shows however, that this cluster is rather heterogeneous and is comprised of trajectories that turn east early and ones that are led around the low pressure system to the west counter-clockwise. As those trajectories likely also experience different processes altering their properties, this cluster is more difficult to interpret than the others, which are more homogeneous. Additionally, these trajectories mainly reside over the Gulf of Biscay and the Mediterranean, therefore, they are less important to interpret in the context of the AD's consequences for central Europe.

The cluster paths and the development of the thermodynamic variables along the path are similar for the initialization on 16 June 2022, and C2 (blue) also remains for the initialization on 17 June 2022. The other clusters are less clear in this case as their travelling time is so short. For the initialization on 18 June 2022 the trajectories are too few and too short to yield meaningful clusters. See Supplementary Material.

## 3.4 Consequences for the local weather in the target region

Warm air advection aloft can suppress cloud formation by confining the local convection to a shallow layer below, hence ADs should be associated with cloud free conditions in the target region. Indeed, large parts of the area covered by the AD are cloud free or only covered by high clouds during the entire study period (not shown here). Medium and low clouds preceding the 800 hPa cold front indicate the region where the AD rises up on the colder air mass at its northwestern edge. It becomes apparent from Fig. 1 that the 800 hPa cold front approaches the AD from the northwest, but only catches up (in parts) with the AD at this level by the end of the case study. This is in accordance with the findings by Dahl and Fischer (2016), who find a similar behaviour in their 3-year composite analysis of EMLs with convergence lines.

Fig. 4 shows vertical cross-sections along 47.5° N (A–B), for 19 June 2022 00:00 UTC, as marked with a red line in Figs. 1 and 3. The AD (grey contour/grid) covers large parts of the lower and middle troposphere, resides higher aloft at its edges and even comes as far down as the surface in its centre, hence penetrating the local BL during the day (not shown here, but discussed in more detail in Sect. 3.4.2). It is not well-mixed in terms of potential temperature (Fig. 4a) or total water content (not shown here). Hence, the AD does not classify as an EML and could not be identified from vertical profiles in the target region, which highlights the necessity of the direct detection method presented in this study for the analysis of ADs. The cold front is clearly visible in the potential temperature in both cross-sections (marked in blue). The horizontal temperature gradient at the western edge of the AD (Fig. 4a) is comparable in strength to the one at the cold front. Even more pronounced it is seen in the equivalent potential temperature due to increased humidity (Fig 4b,c). Similarly, the horizontal and vertical gradients in potential (and equivalent potential) temperature at the northern edge of the AD are of similar magnitude as those at the cold front (not shown here). At the southern edge the gradients are also visible, but weaker. Hence, the lateral edges of the AD are strongly baroclinic zones.

It was suggested that the presence of EMLs can lead to heat waves (e.g. Cordeira et al., 2017). The increased near-surface temperatures seen in Fig. 3 (2 m temperature anomalies compared to 1992–2021) are further discussed in Sect. 3.4.1. The formation of thunderstorms is also influenced by the presence or absence of a lid and is further discussed in Sect. 3.4.3.

### 3.4.1 High near-surface temperatures

In large parts of Europe, the surface temperatures were exceptionally high during the AD event presented here (e.g. Imbery et al., 2022, and Fig. 3). It was proposed that the warm air of EMLs aloft form capping inversions due to their high potential temperatures, which prevent the local BL from growing and reduce vertical mixing (e.g. Carlson and Ludlam, 1968; Carlson et al., 1983; Farrell and Carlson, 1989; Cordeira et al., 2017). Especially under clear-sky conditions, this allows the surface

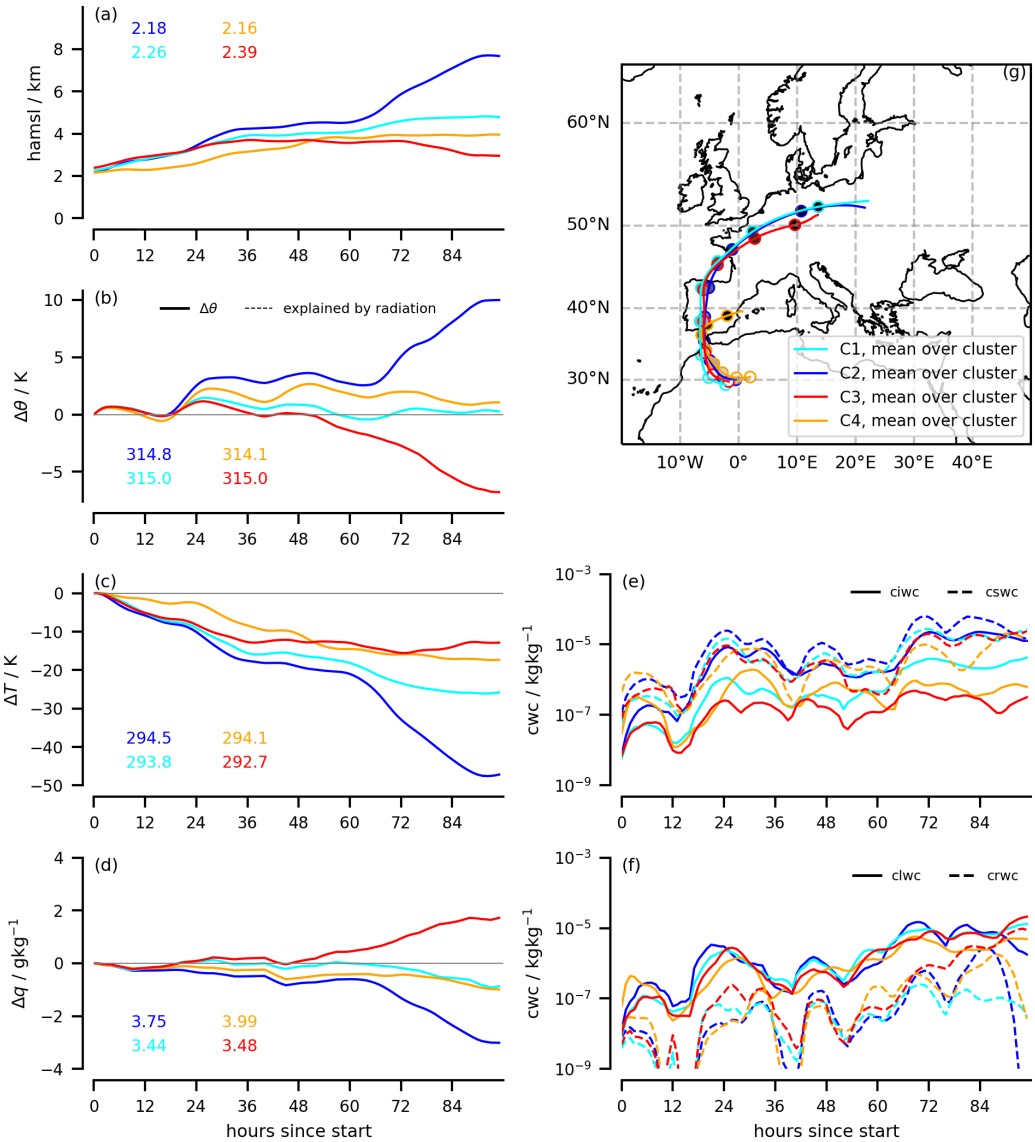

**Figure 2.** Time series of the thermodynamic variables (panels (a–f)) and map of the cluster paths (g) for the cluster mean for C1 (cyan), C2 (blue), C3 (red), and C4 (orange). The mean is calculated in the trajectory relative time frame, the x-axis of the time series shows hours since the trajectory initialization. Panel (a): height above mean sea level ($hamsl$). Panel (b): Difference in potential temperature ($\theta$) since initialization and cumulative mean temperature tendency due to short- and longwave radiation (dashed). Panel (c): Difference in temperature ($T$) since initialization. Panel (d): Difference in specific water content ($q$). Panel (e): cloud ice water content ($ciwc$, solid) and cloud snow water content ($cswc$, dashed). Panel (f): cloud liquid water content ($clwc$, solid) and cloud rain water content ($crwc$, dashed). Panel (g) shows a map of the mean trajectory path. Dots mark every 12th one-hour time step (which corresponds to the x-ticks in the other panels), the colour gradient of the dots represents progression in time, with white being the time of initialization. The coloured numbers is panels (a–d) denote the initial value of the respective variables and clusters in km (a), K (b,c) and gkg$^{-1}$ (d). Panels (e–f) have a logarithmic scale on the y-axis.

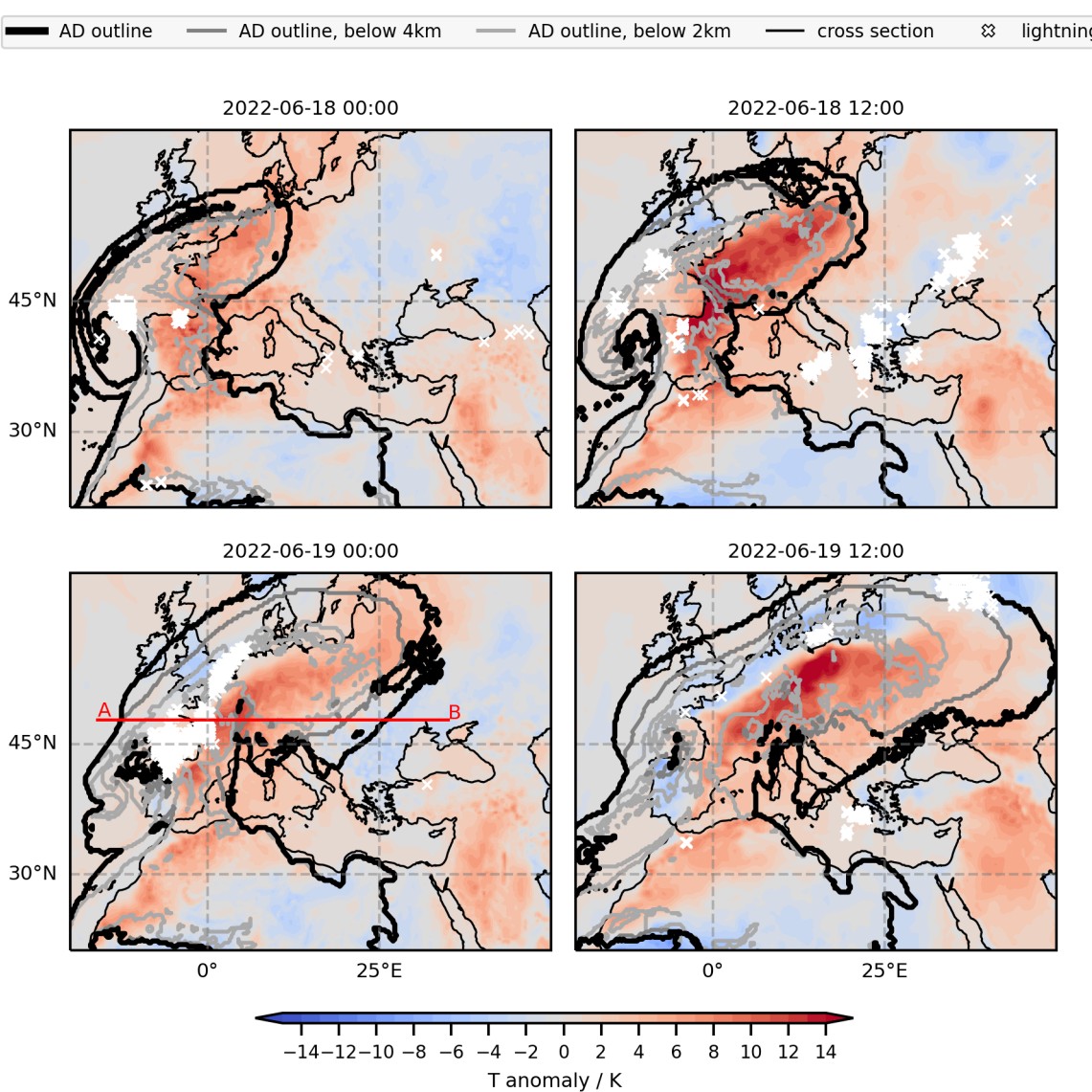

**Figure 3.** Map showing the spatial extent of the AD and the 2 m temperature anomaly with respect to the 30 year period 1992–2021 at 00:00 UTC and 12:00 UTC on 18 and 19 June 2022. The entire AD is outlined in black, outlines of the AD cells up to 4 and 2 km, respectively, are marked in grey. White crosses mark locations where lightning occurred during the hour before and after, 2 m temperature anomalies are coloured with 2 K spacing. Red line (A–B) marks the location of the cross-section depicted in Fig. 4, as in Fig. 1.

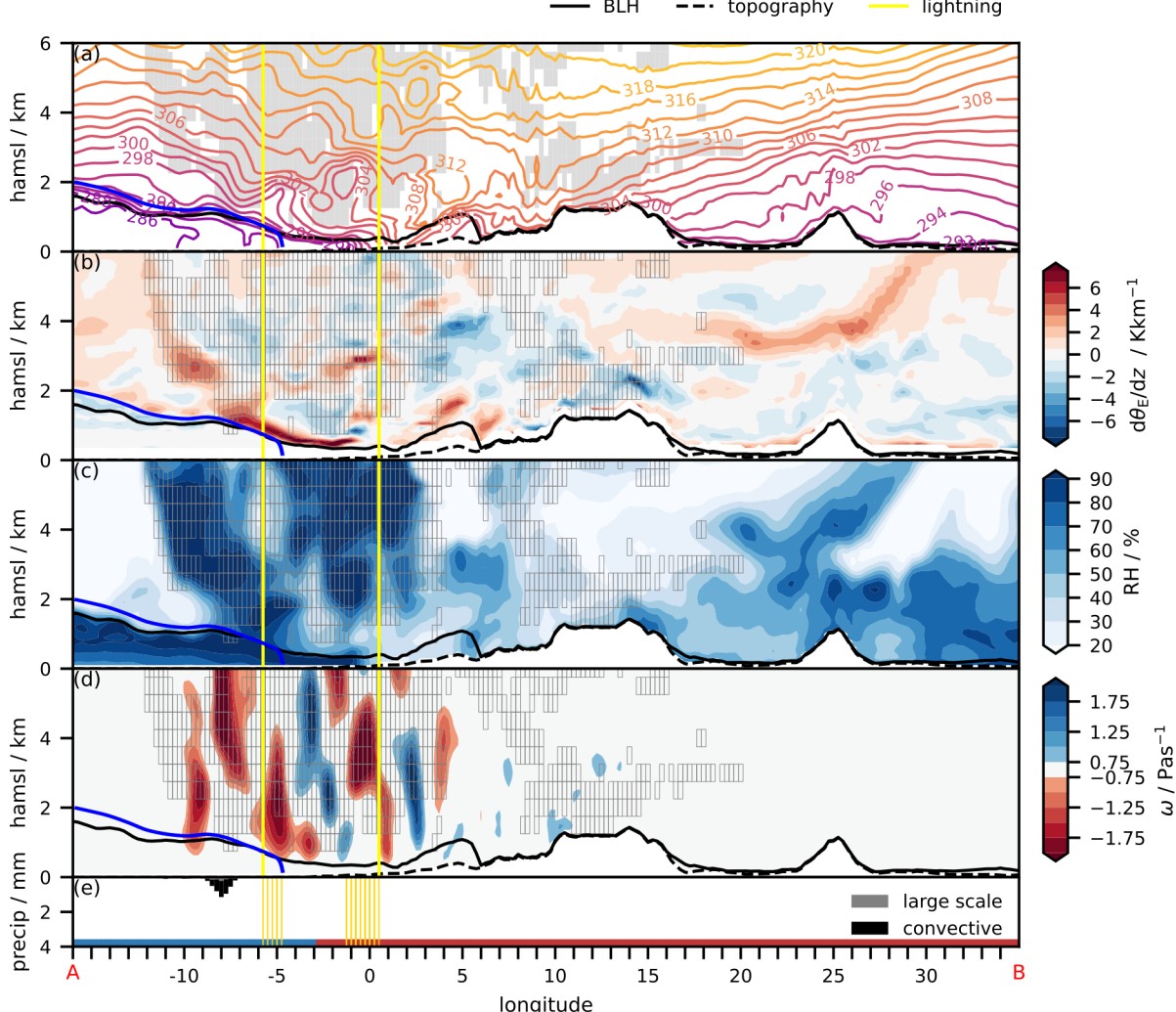

**Figure 4.** Vertical cross-section along 47.5° N, at 19 June 2022 00:00 UTC (16° W to 35° E, 47.5° N, as denoted by the red line (A–B) in lower left panel in Fig. 1). Shown are (a) the potential temperature in K, (b) the equivalent potential temperature gradient in Kkm$^{-1}$, (c) the relative humidity in %, (d) the vertical wind component in Pas$^{-1}$, and (e) the accumulated large scale (gray) and convective (black) precipitation within the previous hour in mm. In panels (a–d), the solid black line denotes the BLH, the dashed black line the model topography, and the vertical yellow lines denote the range in which lightning occurred within a 2 h time window centred at 00:00 UTC and a 1° latitude band centred at 47.5° N. All lightning locations within this range are shown in yellow in panel (e). The region occupied by AD air is marked in grey (shading in (a), grid in (b, c, d)). The cold front is denoted in blue. Land and ocean surfaces are marked along the x-axis in brown and blue, respectively.

temperatures, equivalent potential temperatures, and CAPE to rise to exceptionally high values and lead to heat waves (Cordeira et al., 2017).

The AD may form a lid when the lowest AD-cell in a column lies just above the local BL. At 19 June 2022 12 UTC, 15 % of all the AD-columns north of 37° N have a "lid", identified by the centre of the lowest AD-cell in the column being within ±500 m of the ERA5 BLH. However, the lid was not present for long enough to cause the high near-surface temperatures.

Only in 0.03 % of all the AD-columns north of 37° N a lid was present for longer than 24 h. These columns are in the vicinity of the occluding front in the Gulf of Biscay and over the ocean.

Other possible explanations for the high surface temperatures are advection from warmer regions, subsidence heating, or diabatic processes (also investigated from a Lagrangian perspective by e.g. Bieli et al., 2015; Zschenderlein et al., 2019; Papritz and Röthlisberger, 2023; Röthlisberger and Papritz, 2023; de Villiers, 2020). It was also suggested that land-atmosphere

feedbacks and overnight heat storage in an unusually deep residual layer may intensify heatwaves (Miralles et al., 2014). To investigate the involved processes in this case more closely, back-trajectories from the BL in two regions in eastern Germany and southwestern France which experienced exceptionally high temperatures (Imbery et al., 2022) were calculated (see Fig. 5). As for the forward trajectories, the trajectories are initiated from the defined region, at a resolution of 5 km in the horizontal and 10 hPa in the vertical between 1100 and 400 hPa. Only those that are initially below the local BLH are then used for further

analysis. Some of the back-trajectories from the local BL in those regions, indeed originate in western North Africa and are therefore part of the AD, which penetrated the local BL (see discussion in Sect. 3.4.2). A larger part of the back-trajectories originates over the Atlantic and travels across central Europe, where it subsides. Temperatures increase due to adiabatic heating during the descent (similar situation was found for a Spanish Plume event in 2019 by de Villiers, 2020). Hence, the analysis of the AD event in June 2022 supports the hypothesis that ADs co-occur with anomalously high near-surface temperatures. We

find that not the previously suggested mechanism of a lid, but subsidence heating was dominant in this case. Whether this is caused or facilitated by the presence of the AD, or whether the co-occurrence is coincidental cannot be determined with one case study and will be subject of future research.

### 3.4.2  Penetration of AD-air into the target region's boundary layer

In the centre of the AD air mass, AD air penetrates the local BL, which may be surprising, considering that the AD air can be

expected to be warmer and dryer than the local BL air. The impact of the trajectories that enter the local BL on the clustering analysis is small, they constitute only 0.8 % of the 8.7 million trajectories in total. For the AD air to be entrained into the local BL, they must have a similar potential temperature. As already mentioned, the near-surface temperature was already unusually high. Additionally, the trajectories that did enter the local BL by 19 June 2022, 12:00 UTC, have cooled considerably, while the trajectories that end up above the local BL have warmed (for trajectories initiated on 15 June 2022 the cooling is about

6 K and the warming about 2 K on average). One reason for this different behaviour is that the trajectories entering the local BL experience more radiative cooling along the way on average (not shown here). The development of the specific water contents indicates that another reason for the difference is latent heating. While the trajectories that end up above the local boundary layer seem to form condensate which causes latent heat, the trajectories that enter the local boundary layer cool due

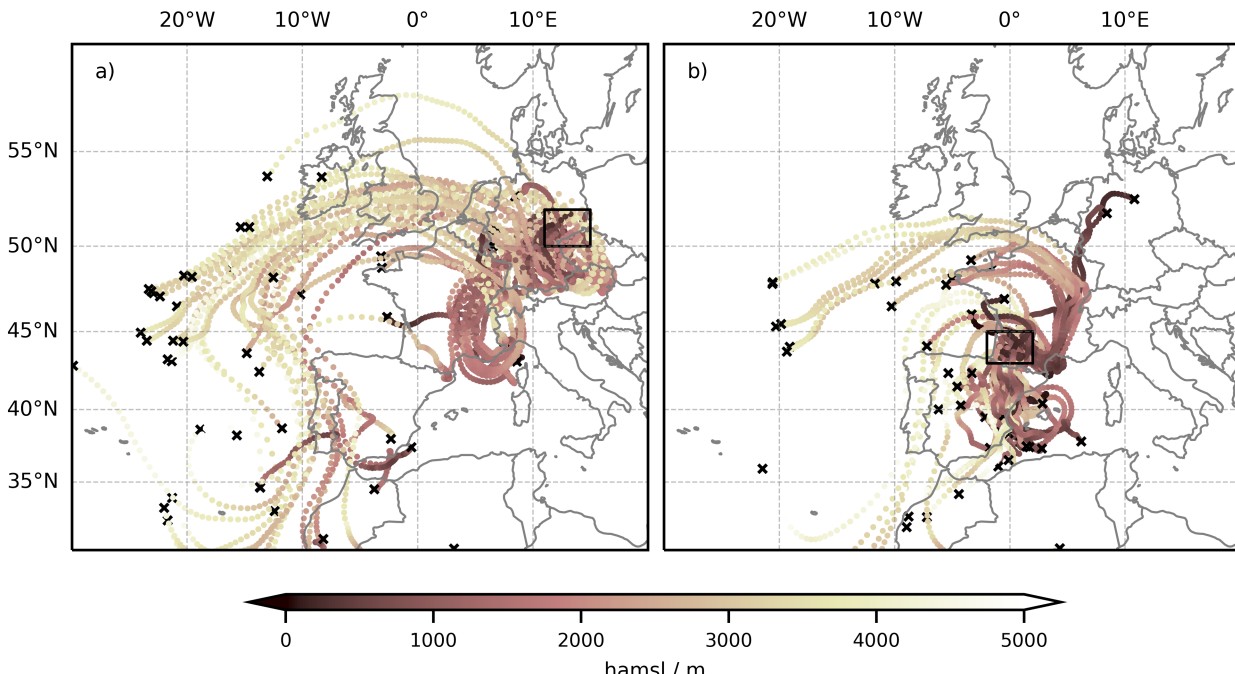

**Figure 5.** Back-trajectories from areas with elevated surface temperatures. A random subset of 20 back-trajectories that are within the local BL at the time and location of initiation is shown. Colours indicate the height above mean sea level ($hamsl$) in metres. Black crosses mark the position at 15 June 2022 00:00 UTC. Panel (a) shows the back-trajectories started at 19 June 2022 12:00 UTC from the box in eastern Germany marked in black. Panel (b) shows the back-trajectories started at 18 June 2022 12:00 UTC from the box in southern France marked in black.

to evaporation, likely of precipitation falling through from above (not shown here). Hence, high near-surface temperatures in the local BL and cooling of the trajectories due to radiation and re-evaporation of precipitation make it possible for AD air to enter the local BL.

### 3.4.3 Thunderstorms

It has been described for EMLs that the capping inversion due to the warm, well-mixed layer aloft suppresses thunderstorm formation in the centre of the EML, while thunderstorms tend to break out violently along its edges (often in the vicinity of cold fronts or dry-lines; Carlson and Ludlam, 1968; Carlson et al., 1983; Farrell and Carlson, 1989; Keyser and Carlson, 1984; Dahl and Fischer, 2016). The analysis of the case presented here suggests that this behaviour is similar for the more general case of an AD. During 18 and 19 June 2022, the majority of the region covered by the AD experiences no lightning, except for a line parallel to the cold front, where lightning occurs during the early hours of 19 June 2022 (Fig. 1, bottom right). This implies that thunderstorms are suppressed, probably due to a combination of subsidence in the high pressure system and the presence of the (warm) AD aloft. The thunderstorms that do occur are close to the edge of the 800 hPa AD layer. Only in the

early evenings of 16 and 17 June 2022, there is strong lightning activity over the Iberian peninsula (not shown here) which is likely due to typical summer heat thunderstorms which are not suppressed by the warm AD air aloft, although it has already been there since the early hours of 16 June 2022. During 18 and 19 June 2022, there is also some lightning activity over Spain and the Gulf of Biscay, in the vicinity of the occluding low pressure system (Fig. 1).

Carlson and Ludlam (1968) and Carlson et al. (1983) argue that thunderstorms are initiated along the edge of the EML, because heated, moistened air from underneath the lid *underruns* it. It then reaches the edge, where the lid is higher and weaker, so that the constraint is weaker, convection can penetrate and thunderstorms can break out. Similarly, along the edge, surface heating by insolation can suffice to overcome the lid. Using the Sawyer-Eliasson equation and quasi-geostrophic theory, Keyser and Carlson (1984) conclude that due to the confluence at the midlevel baroclinic zone, a thermally direct circulation

develops there, while anti-cyclonic shear induces a cell of thermally indirect circulation in the upper parts of the EML. These circulations act together to induce a branch of rising motion at the midlevel baroclinic zone. They argue, however, that it should be too weak to cause thunderstorm outbreak and rather supports other mechanisms by further weakening the lid. Similarly, Dahl and Fischer (2016), who used Q-vector analysis, found that in the warm season, when EML are influenced by a cyclone's low-level wind field, a convergence line forms along the western edge of the EML, east of the 850 hPa cold front, facilitating

lifting and the initialization of thunderstorms. Also Cordeira et al. (2017) argue that the presence of an EML can support strong thunderstorms, if meso–synoptic-scale lifting mechanisms initiate convection.

    A closer analysis of the vertical cross-section in Fig. 4 can give some insight into the processes involved in the thunderstorm formation in this case. There are two accumulations of lightning along the latitudinal cross-section in Fig. 4. The first is located between 4.75° and 5.75° W. This coincides with the location of the surface front, at which the air mass above is lifted (red

shading in Fig. 4d). Above 4 km, the potential temperature increases slowly with height (Fig 4a), while the equivalent potential temperature decreases with height (Fig. 4b), which indicates potential instability. As the approach of the front lifts the air mass aloft, this potential instability is released and thunderstorms break out. Relative humidity close to 100 % (Fig. 4c) is also an indicator of the clouds that form in response to the convection.

    The second accumulation of lightning is located between 1.25° W and 0.5° E . This region is close to the eastern edge of the

800 hPa AD layer (see Figs. 1 and 4) and has a very strong horizontal temperature gradient at the surface, preceding the surface cold front. Here, air converges (not shown here), there is rising motion present associated with a thermally direct circulation, and above 3 km altitude the equivalent potential temperature decreases. Hence, the lifting can release potential instability. This supports the argument by Keyser and Carlson (1984) and Dahl and Fischer (2016), although there seems to be no thermally indirect circulation that contributes to the rising motion. Around 8° W, in the vicinity of the western edge of the AD at 800 hPa,

there is another region with strong ascending motion. Here, this rising branch is associated with a thermally indirect cell. While there was no lightning recorded in this region, ERA5 shows some convective precipitation, which indicates that convection was initiated, although not producing any thunderstorms.

    A longitudinal cross-section 3° E, 42° N to 3° E, 59° N, helps understanding the lightning that occurs between 51.5° and 53.25° N (not shown here). This region experiences strong upward motion above 2 km, and is located north of the surface

front, but south of the 800 hPa front and the AD edge at that level. Also here there is a zone of convergence (not shown

here). Centred at the region with strongest upward motion, there is a wider region where ERA5 produces both large scale and convective precipitation in response to the lifting. In this case, there is increased CAPE at the locations of the thunderstorms, which is released. However, the rising motion is not surface-based, which it should be if underrunning were the reason for it.

## 4   Conclusions

In this study we introduced the concept of atmospheric deserts (ADs), which can be seen as a generalization of elevated mixed layers (EMLs), as they are air masses originating in the dry, hot, convective boundary layers (BLs) of semi-arid, desert, subtropical and/or elevated source regions that are advected to an often cooler, moister target region. Since they progressively lose their distinct characteristics during the advection over hundreds to thousands of kilometers, they cannot be detected based on their thermodynamic properties in the target region.

We introduced a direct detection method, tracing the air mass directly from source to target using Lagrangian trajectories. This allows a detailed and high-resolution analysis of these air masses and their developments during the advection. Different trajectories travel along different paths and also experience different diabatic processes that change their properties. A clustering of the trajectories is used to analyze the typical pathways.

A case study for 15–19 June 2022 is used as an example to explain the involved processes. The AD in this case travels
across large parts of southern and central Europe during the duration of the case study and occupies many layers in the vertical between the surface and 13 km, extending further to the surface in its centre, and residing higher aloft at its edges. The 800 hPa cold front approaches the AD from the north west, but only catches up with it by the end of the case study.

Four different trajectory pathways were identified. Of the four, only one cluster behaves like one could expect of a typical EML: It rises slightly over the colder, local air mass, while almost conserving its potential temperature and water vapour
mixing ratio and, therefore, the well-mixed properties of the source region's CBL. Diabatic processes, however, modify the properties of the trajectories in the other clusters. One cluster rises much higher, thereby cooling adiabatically, which induces condensation. Latent heat causes the potential temperature in this cluster to rise and condensate precipitates out. Another cluster experiences a descent (after an initial ascent together with the other clusters). Meanwhile, its specific water vapour content increases and its potential temperature decreases. This is partly due to radiative cooling, and partly due to re-evaporation of
precipitation falling through from above. Also, mixing with the cooler, moister local air can be a reason for the cooling and moistening. A fourth cluster is distinguished mainly by the deviation in its geographical location, but more difficult to interpret due to its heterogeneity and less interesting because it does not travel across central Europe.

ADs can have similar consequences for the weather in the target region as were described for EML: Europe experienced positive near-surface temperature anomalies during this AD event, and thunderstorm initialization was limited to a narrow
line. In this case study, however, the AD did not form a lid for long enough for heat to build up underneath. An analysis of back-trajectories from heat-affected regions indicates that subsidence heating contributes to the increase of near-surface temperatures. When the near-surface temperatures are increased, it is also possible for AD air that is cooled during the advection

to penetrate the local BL. This happens for a small percentage of the trajectories that experience strong cooling due to a combination of radiation and evaporation.

Thunderstorms broke out in the vicinity of the occluding low pressure system over the Gulf of Biscay, and along a line, parallel to the edge of the AD and the approaching cold front. Further analysis showed that some of the thunderstorms are initiated in the close vicinity of the surface cold front, when it lifts the potentially unstable, overlying air mass. Another region of ascending motion and thunderstorms was found between the surface and the 800 hPa front, close to the edge of the AD at the 800 hPa level. Several processes were suggested to cause the outbreak of thunderstorms along the edges of such an air mass. The present case study supported some of the arguments from literature, namely ascending motion due to confluence at the edge, but the data is not sufficient to find more causal relations about why the thunderstorms break out exactly where they do.

It may be interesting to compare ADs with atmospheric rivers, a phenomenon potentially causing severe rainfall (e.g. Ralph et al., 2018), as they are both larger scale atmospheric phenomena leading to extreme weather in their target regions. However, while the name "atmospheric deserts" is inspired by "atmospheric rivers", the phenomena are not really opposites. Atmospheric rivers are defined by their water vapour footprint, but atmospheric deserts are defined solely by their source region.

The direct detection method introduced in this study makes it possible to actively trace air masses from their source to their target region. It is therefore possible to study more general cases than the EMLs studied with indirect detection methods in the previous literature. Additionally, tracing the air mass allows to investigate processes that modify the air mass along its way and thereby improve our understanding of the processes involved. The high number of trajectories itself is a novelty, typically only tens to hundreds of trajectories are analysed (e.g. Zschenderlein et al., 2019; Sprenger and Wernli, 2015), and usually to confirm the origin of an air mass rather than to identify it. Some studies use thousands of trajectories to identify the origin of high temperatures (Papritz and Röthlisberger, 2023; Bieli et al., 2015). In contrast, we use several millions of trajectories to identify the air mass which renders our results very robust. For example, we obtain qualitatively identical results when – in addition to the 8.7 million trajectories – we analyse another 28 million trajectories that originate in a smoothed nocturnal BL. Identifying a cell as an AD-cell based on the presence of a single trajectory does not mean that it is dominated by AD air. However, it is rarely the case that only one trajectory is present in a cell, hence misidentification is not a big risk. This way, the identification method also remains applicable in cases where fewer trajectories were initiated. Therefore, the method is also suitable for comprehensive climatological analyses of properties of atmospheric deserts and the processes involved in modifying them from their source region to their target region.

The case study presented in this work helped to understand the processes modifying AD air along the way and the influence of the AD on local weather in the target region. Future work is planned to generalize the results presented here and investigate the processes related to heat waves and thunderstorm formation during ADs in more detail, based on longer time series of AD events. It will be interesting to study whether the heat buildup under the lid really plays a minor role compared to other mechanisms leading to high near-surface temperatures in the presence of an AD, or whether this was case-specific.

*Code availability.* The code used to calculate the trajectories and results and plots presented here can be found at: https://doi.org/10.5281/ zenodo.12663678

*Data availability.* ERA5 data is freely available at the Copernicus Climate Change Service (C3S) Climate Data Store (Hersbach et al., 2023). The results contain modified Copernicus Climate Change Service information 2020. Neither the European Commission nor ECMWF

is responsible for any use that may be made of the Copernicus information or data it contains. Lightning data from Blitzortung.org is available as participant of measurement network. The LAGRANTO is available from: Sprenger and Wernli (2015).

*Author contributions.* GM and AZ acquired the funding for this project. FF conducted the calculations, the analysis, and wrote the manuscript under supervision by GM and AZ, with the support of IS and RS. IS acquired the data and RS supported in software development. GM, AZ, IS, and RS reviewed the manuscript prepared by FF.

*Competing interests.* The authors declare that they have no conflict of interest.

*Acknowledgements.* We thank Deborah Morgenstern for setting up LAGRANTO and running the first explorative trajectory calculations. We thank all colleagues who were involved in discussions. The computational results presented have been achieved [in part] using the Vienna Scientific Cluster (VSC). This work of Fiona Fix was funded by the Austrian Science Fund (FWF, grant no. P35780).

**Appendix A**

Fig. A1 shows cluster C2 (blue) as in Fig. 2, but together with the cluster median, the interquartile range and three individual trajectories from that cluster. From panel (a) it becomes obvious that the individual trajectories experience the jump in altitude at very different times and the shape of the mean and median might that there are two dominant times for this to happen: around 24 h and around 80 h. The changes in $\theta$, $T$, and $q$ are consistent with this behaviour (panels (b)–(d)). From panels (e) to (f) it becomes apparent, that the mean and median in the cloud water content variables differ considerably. This can be explained

by the large spread between the individual trajectories and the skewed distribution. Since each trajectory shows peaks in those variables at slightly different times, there are many trajectories with little cloud water content at all times, and only few with very high values. Panel (c) shows that on average the trajectory temperature only sinks below the freezing level (marked by horizontal line) around hour 60. The fact that there is frozen cloud water content present even before this (panel (e)) is due to the same reasons as the discrepancy in the mean and median. If one pays attention to the three individual trajectories, it can be

seen that they do have physically consistent time series in all variables. Hence, it should be kept in mind that the cluster mean

(or median) do not represent a physically consistent trajectory. They are, however, still useful to discuss average behaviour of the trajectories in the cluster.

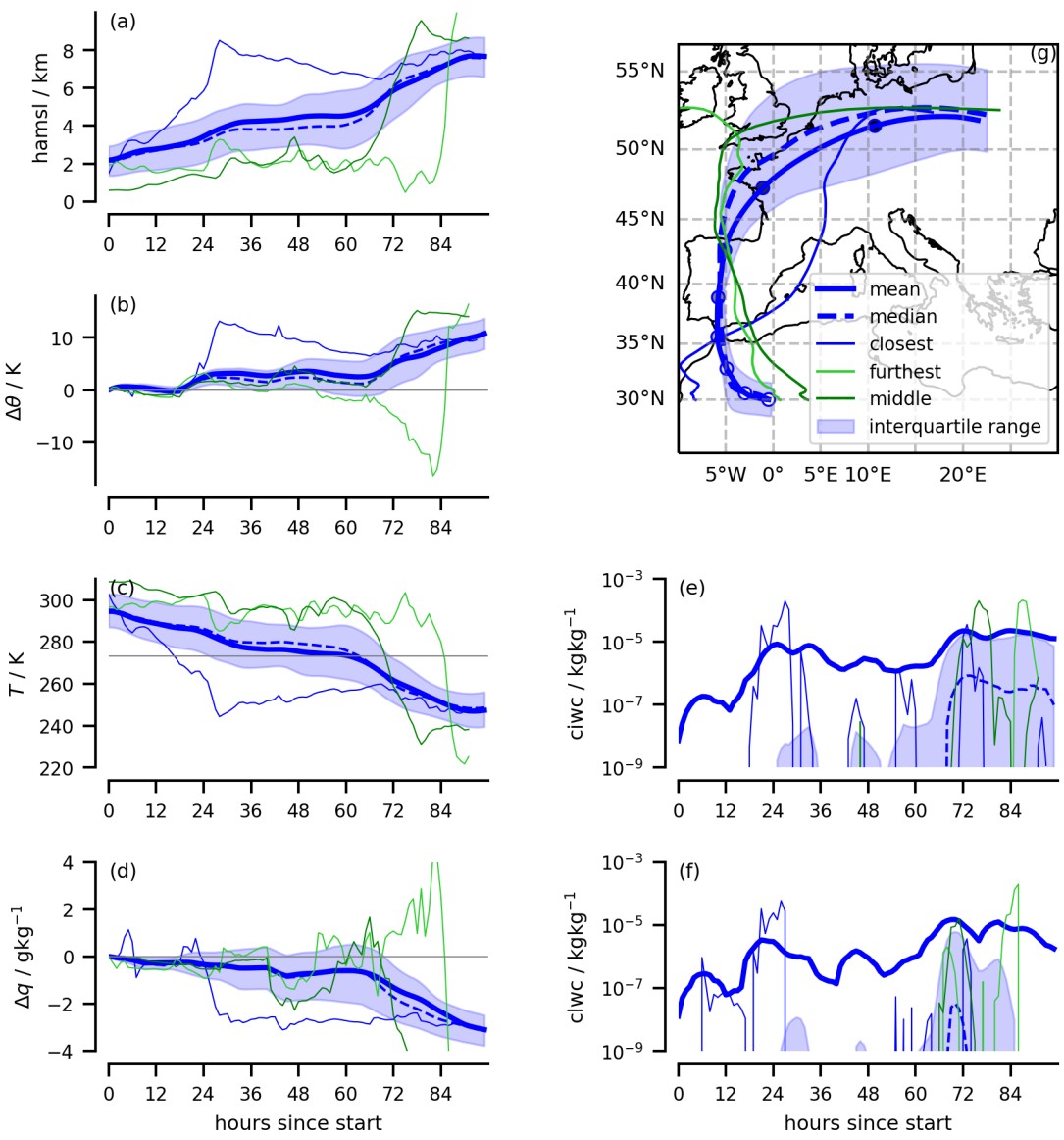

**Figure A1.** Detailed depiction of the blue cluster (C2) of the trajectories started on 15 June 2022, as in Fig. 2. The thick blue line denotes the cluster mean, the dashed blue line the cluster median, and the shaded region marks the interquartile range. The thin blue line marks the trajectory that is closest to the cluster centroid, the light green one the one that is furthest from the centroid, and the dark green is a trajectory with a medium distance to the cluster centroid. Panel (a): height above mean sea level ($hamsl$). Panel (b): Difference in potential temperature ($\theta$) since initialization. Panel (c) Temperature ($T$). Panel (d): Difference in specific water content ($q$) since initialization. Panel (e): cloud ice water content ($ciwc$). Panel (f): cloud liquid water content ($clwc$). Panel (g): Map of the trajectory pathways. Panels (e–f) have a logarithmic scale on the y-axis.

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
