# Peer review of "Detection and Consequences of Atmospheric Deserts: Insights from a Case Study"

_EGUsphere, 2024_

## Referee Comment (RC1)

**Broad context and summary of the key results of the present study:**

Warm and very dry air masses emanating from convective boundary layers over arid regions can be advected to remote regions, with a possibly profound impact on the weather there.
This phenomenon might be important in European summers, as the advection of such air masses may be associated with severe weather phenomena such as heatwaves and thunderstorms.
The authors base their study on previous work focusing on so-called elevated mixed layers (EML). By allowing for a gradual loss of the defining properties of EML during the advection towards Europe, the authors introduce a more generalized concept called atmospheric deserts (AD), which focuses mostly on the origin of the air mass. The robust detection of such air masses is based on a Lagrangian approach, for which a very high number forward trajectories are computed from the typical North African source region.
Using a case study of June 2022, the authors demonstrate the results of their Lagrangian detection algorithm. By applying k-means clustering, it is shown that only a certain fraction of the considered air masses retain their original well-mixed properties, while a majority is affected more or less strongly by diabatic processes along their way to Europe.
Finally, it is discussed in which ways AD exert an influence on the local weather in Europe. Within the centre of ADs,  heatwave formation might be fostered due to convection suppression and build-up of heat due to the AD-induced capping inversion. In contrast, along the boundaries of ADs thunderstorm activity might be increased due to a combination of dynamical and thermodynamical mechanisms.

**General comments:**

The present study provides a well-motivated and intriguing investigation into the concept of atmospheric deserts. It builds upon existing literature about elevated mixed layers, but provides a novel perspective by focusing on Europe and using a Lagrangian framework. While the emphasis of this paper is on introducing the methods of AD detection, the inclusion of a case study also provides a first glimpse at the potential importance of such air masses for European summer weather.

Overall, I am pleased with the quality of this work. The paper is well-written and mostly easy to understand. For the most part, the methods used are sound and well-suited for the objectives.
Both the presentation and the interpretation of the results easily meet the standard of this journal. I am particularly happy about how the potential impacts of ADs on the local weather during the case study are really communicated carefully, always making clear that other processes than the presence of ADs might have been more important for heat build-up or thunderstorm initiation.

In conclusion, I am therefore positive that this work is suitable for publication in WCD.
Below I have a few comments and questions mostly regarding the methods, which can hopefully be addressed easily. If that is the case, this publication can be accepted after some minor or possibly major revision.

**Specific comments:**

L58-60:
I was wondering how strongly your results depend on the choice that a grid cell is already defined as an AD grid cell when only a single forward trajectory finds its way to the given grid point. I would assume

that the fact that the end point of one forward trajectory lies in a grid cell probably does not necessarily mean that the local air mass is then strongly characterized by the AD air mass properties.
Maybe this could be tested by initiating a certain number of backward trajectories from an AD grid point? If a majority of these backward trajectories can be traced back to the African source region, then it is safe to say that the local air mass is indeed strongly composed of the AD air.
In the end it is also simply a matter of perspective. In your article, you are strongly focused on the investigation of the air masses emanating from dry convective boundary layers, their properties and the changes they undergo during the transport. From that perspective it makes perfect sense to use forward trajectories. So I do not view this as a potential major problem for your article.

L68-75:
As far as I understand you want to include the nocturnal residual layer air masses for your Lagrangian approach.  However, it is in my opinion not fully appropriate to use this kind of a smoothing algorithm because the residual layer might also be of similar height than the daytime convective boundary layer; it might also develop in a certain way over night depending on environmental parameters. The smoothing is unlikely to fully reflect this behavior.  At the same time, you maybe would also want to exclude the lowermost layers, which will feature a stable nocturnal boundary layer forming under strong radiative cooling.
I would suggest to test whether your results would substantially change if you would only use day-time data for which the use of ERA5's boundary layer height might be suitable to estimate the height of the convective boundary layer.

L80-83:
Although I think that I understand how the clustering is performed, it would be beneficial to state a bit more clearly that you are using a multivariate clustering approach. I assume that the data points are clustered within a 11-dimensional space, in which each dimension reflects one of the 11 standardized variables? (such that any of the variables have the same weight)

L245-248:
At least in my opinion, this short paragraph appears to be of minor importance compared to earlier paragraphs. Looking at the plots, I also had problems to identify the air mass that is found between the near-surface air and the overlying AD air mass. Therefore I would suggest to rephrase or remove this short paragraph.

L275-280:
I would suggest rewording these three sentences. In their current form, they confused me somewhat because I had difficulty understanding which kind of diabatic processes predominate over other diabatic processes.

**Minor comments:**

L19:
In the parenthesis, the listing "economic, ecologic, …" should be removed or replaced by a formulation such as "can cause severe economic and ecologic damage"

L46 and elsewhere:
I would replace "thunderstorm eruption" and "thunderstorms erupt" by some other formulations such as "thunderstorm outbreak" or "thunderstorm initiation" and "thunderstorms are initiated.." or "thunderstorms occur.."

L144:
typo: 18 June 2002 should be 2022

L313:
I think it should say "western edge" instead of "eastern edge" if I am not mistaken

L346:
"Diabatic process**es**"

---

## Author Comment (AC1)

Dear reviewer,

thank you for your positive feedback and the constructive comments. In the following, we address the comments individually.

**Specific Comments:**

*L58-60: I was wondering how strongly your results depend on the choice that a grid cell is already defined as an AD grid cell when only a single forward trajectory finds its way to the given grid point. I would assume that the fact that the end point of one forward trajectory lies in a grid cell probably does not necessarily mean that the local air mass is then strongly characterized by the AD air mass properties. Maybe this could be tested by initiating a certain number of backward trajectories from an AD grid point? If a majority of these backward trajectories can be traced back to the African source region, then it is safe to say that the local air mass is indeed strongly composed of the AD air.*

Thank you for this comment. You have a valid point. However, calculating the trajectories is costly in computing time and data output, and we want to make the detection method feasible to use on long time series of data. Therefore, we want to avoid this extra step.

Additionally, the number of trajectories that end up in one cell is not only dependent on whether this cell is actually dominated by AD air, but also on how many trajectories were started in the first place. As we aim to make the detection method described in this paper computationally feasible even when run over longer times (i.e. years), the total number of initiated trajectories has to be further reduced. Also a different handling of the nighttime CBL (see answer to your second comment) might reduce the total number of trajectories drastically. Hence, using the presence of at least one trajectory in a cell as an indicator is a useful approach.

One approach to embrace your comment without increasing the computational needs would be to set a higher minimum number required for the identification of an AD cell. As this decision influences the definition of the AD edge, this is important for all the results and discussion related to Figures 1, 4 and 5 in the manuscript.

As can be seen in Figures 1 to 3 included below, in the regions where the location of the AD edge is critical to the discussion, there are many trajectories in the cells, so that a higher minimum number would not change the identification. Figure 4 (included below) shows the maximal extent of the AD, i.e. the colouring stands for the amount of trajectories in the respective columns. It becomes clear again, that most columns have a very high number of trajectories in them, hence not many are classified just because there is one trajectory in them.

We hope, that with this answer we could convince you that labelling a cell as an AD cell when it contains at least one trajectory is a useful approach and likely does not misidentify the AD edges drastically.

*L68-75: As far as I understand you want to include the nocturnal residual layer air masses for your Lagrangian approach. However, it is in my opinion not fully appropriate to use this kind of a smoothing algorithm because the residual layer might also be of similar height than the daytime convective boundary layer; it might also develop in a certain way over night depending on environmental parameters. The smoothing is unlikely to fully reflect this behavior.*

You are certainly correct, our smoothing approach cannot reflect the true residual layer. The idea was to use a smoothed BLH that still is lower during the night, so we rather under- than overestimate the true residual layer.

*I would suggest to test whether your results would substantially change if you would only use day-time data for which the use of ERA5's boundary layer height might be suitable to estimate the height of the convective boundary layer..*

We recalculated the results using only trajectories that were initiated during the daytime (1-5pm, including 1 and 5pm) and from below the ERA5 BLH. As the results do not substantially change, we have incorporated this in the manuscript now. Note, however, that this reduces the total number of trajectories significantly and we therefore again have to assume that an AD cell can be identified as such based on a single trajectory (see comment before).

*L80-83: Although I think that I understand how the clustering is performed, it would be beneficial to state a bit more clearly that you are using a multivariate clustering approach. I assume that the data points are clustered within a 11-dimensional space, in which each dimension reflects one of the 11 standardized variables? (such that any of the variables have the same weight)*

You did understand that correctly and we have added the following sentence to increase clarity: "We employ the 11 variables listed in Tab. 1 and the data driven k-means-clustering method (MacQueen, 1967) to cluster the trajectories. This is a multivariate clustering approach,using the 11-dimensional data after normalizing all variables (zero mean and unit variance) in order to give all of them the same "weight" or "importance" in the clustering."

*L245-248: At least in my opinion, this short paragraph appears to be of minor importance compared to earlier paragraphs. Looking at the plots, I also had problems to identify the air mass that is found between the near-surface air and the overlying AD air mass. Therefore I would suggest to rephrase or remove this short paragraph.*

Indeed, this paragraph is not crucial to the discussion and was therefore removed.

*L275-280: I would suggest rewording these three sentences. In their current form, they confused me somewhat because I had difficulty understanding which kind of diabatic processes predominate over other diabatic processes.*

We have rephrased these sentences to: "Additionally, the trajectories that did enter the local BL by 19 June 2022, 12:00 UTC, have cooled considerably, while the trajectories that end up above the local BL have warmed (for trajectories initiated on 15 June 2022 the cooling is about 6 K and the warming about 2 K on average). One reason for this different behaviour is that the trajectories entering the local BL experience less radiative cooling along the way on average (not shown here). The development of the specific water contents indicates that another reason for the difference is latent heating. While the trajectories that end up above the local boundary layer seem to form condensate which causes latent heat, the trajectories that enter the local boundary layer cool due to evaporation, likely of precipitation falling through from above (not shown here)."

**Minor Comments:**

We incorporated the minor comments directly in the text.

*L313: I think it should say "western edge" instead of "eastern edge" if I am not mistaken*
Thank you for pointing this out to us. Actually, "eastern edge" was correct, but the coordinates were accidentally swapped, which we now corrected. The paragraph before is about the lightning that occurs at the surface front (close to the **western** edge, but this paragraph is about the lightning that occurs close to the **eastern** edge of the AD at the 8800 hPa level. The coordinates were corrected in the text. The content of the discussion was correct, however.

**Additional Comments:**

We found a minor error in the calculation of the 2m temperature anomaly, which we corrected, so an updated version of Figure 3 will be found in the revised version of the manuscript (changes are small and do not influence the discussion).

[Figure]

Figure 1: As Figure 1 in manuscript: AD extent in 800-750 hPa level in shaded contours, maximum extent outlined in black. Fronts and lightning omitted in this case, for better visibility. Colour scale refers to number of trajectories in the respective cell.

[Figure]

Figure 2: As Figure 4 in manuscript: AD air is marked in grey shading in the top 2 panels, the colour scale refers to number of trajectories in the respective grid box.

[Figure]

Figure 3: As Figure 5 in manuscript: AD air is marked in grey shading in the top 2 panels, the colour scale refers to number of trajectories in the respective grid box.

[Figure]

Figure 4: Colour shading for number of trajectories in the respective column, i.e. showing the maximal AD extent (as marked in black outline in 1). Attention: extended colour scale compared to Fig. 1.

---

## Author Comment (AC2)

**Answer to the Editor**

Dear Peter Knippertz,
thank you for your comment and excuse our slow response.

Reviewer 2 criticised that the title raises expectations that are not fulfilled by "only" presenting one case study and suggests adding a climatology of atmospheric deserts. These are very valid concerns, please see below for our detailed answer to the major and main minor comments of reviwer 2. In summary, we can say, that in order to address the mentioned concerns, we change the title of the manuscript and parts of the text, to make clear that we introduce the concept and illustrate it with one case study only. However, we will not incorporate a climatological analysis in this study, as this requires comprehensive analysis and would be beyond the scope of one paper, as you mentioned yourself. To address the remaining major comment, we also include a more detailed discussion about the definition/detection of ADs based on single trajectories, showing that using only single trajectories to identify AD-cells does not lead to a vast mis-identification.

With your consent we are happy to provide a complete answer letter, containing both reviewer's comments, our respective answers and the changes made to the manuscript, together with the reviewed manuscript and a document highlighting the changes made. Since some of the co-authors were out of office and we could only answer to the second review now, we would appreciate an extension of the deadline. We have requested an extension of the deadline by 15 days in the online tool already. However, there is a week-long downtime scheduled next week at the high performance computing facility we work at. Therefore, we would be very glad if you could grant us an extension until the end of October. We will gladly provide all required documents as fast as possible.

**Answer to Reviewer2**

Dear reviewer,
thank you for your feedback and the constructive comments. Below, we address your main comments individually.

**Major Comments:**

*1. In my view, the title of the study promises a little bit too much. When I first read the title, I imagined that the authors were introducing a new concept and showing how this new concept affects weather in the extratropics in general. In reality, the authors "only" performed a case study, which is not negative in itself, but not enough for this title. This brings me to my main critique of this study. Although the case study is certainly very nicely analysed and interesting, the main findings of the study are not too significant. [...]*

Thank you for this comment. We understand how you might expect more of the study, based on its title. To address your concern, we adjust the title to "Detection and Consequences of Atmospheric Deserts: Insights from a Case Study". We adjust the descriptions in the abstract, introduction and where appropriate accordingly, in order to not mislead the readers.

Nevertheless, we highlight that the objective of this study is to introduce the new concept of atmospheric deserts based on the idea that they should be a generalisation of EMLs. We base the conceptual interpretation on literature about EMLs and our physical understanding of the involved processes. The case study in this study is used to explain how the direct detection method is applied and to gain first insights into how ADs might be similar or dissimilar to EMLs.

*2. Climatological analysis: I know that this needs very much computing time. But I would suggest that the study would extremely benefit from that. For a climatological analysis a calculation of trajectories on a much coarser resolution would be sufficient (I think this would be feasible in terms of computation time). [...]*

We do appreciate that a systematic analysis of a climatology of ADs would be beneficial. Such a study is planned for the near future. However, this would be beyond the scope of this paper. The resolution of the trajectories used in this paper is not feasible as it requires a lot of computational resources and produces vast amounts of data. As you suggest, reducing the resolution is a solution, but a careful experiment setup is required, as the identification of the AD air mass is sensitive to the amount of initiated trajectories (also see our answer to your major comment 4). Additionally, ADs can be expected to be a rare phenomenon, which makes the calculation of a smooth climatology challenging. We are planning to make use of gerneralized additive models that have been shown to outperform simply using average cell-counts (i.e. Simon et al. 2017). Furthermore, a climatology of AD occurrence "only" is not sufficient to answer all the questions raised in the manuscript. A thorough analysis of the (co-)occurrence of high near-surface temperatures and thunderstorms is required as well. We also plan to use statistical methods to determine whether the presence of an AD is a

useful predictor for these extreme events. While this is interesting and necessary research, incorporating all this comprehensive analysis in this paper would result in a very extensive paper, which we do not deem beenficial.

*3. Maybe you can discuss the atmospheric desert with the opposite "atmospheric river"? Both can lead to extreme events, one to extreme precipitation and the other one to extreme temperatures? Maybe you can elaborate on this (also this discussion would need a climatological analysis)*

This is indeed an interesting point. Since a climatological analysis is beyond the scope of this study, we cannot discuss this in much detail. However, we will address this in the discussion. The comparison is not very straight forward, however. While the name "atmospheric deserts" is inspired by "atmospheric rivers", the phenomena are not really each others opposite. Atmospheric rivers are defined by their water vapour footprint, but atmospheric deserts are defined solely by their source region.

*4. I find the definition of an atmospheric desert in L60-61 a bit too weak. Is only one trajectory really enough to significantly affect a grid box? Maybe you could perform a sensitivity analysis and elaborate on that.*

Thank you for this comment, you have a valid point. Reviewer 1 raised a similar concern. However, the number of trajectories that reach one cell is not only dependent on whether this cell is dominated by AD air, but also on how many trajectories were initiated. We aim to make this methods computationally feasible for longer time series and climatologies, and therefore have to further reduce the amount of trajectories initiated. In order to address your concern, we have re-plotted Figures 1, 4 and 5 in the manuscript so that they show the actual number of trajectories in the cells (see Figures 1-3 below). Note, that according to the comment of reviewer 1, we also only use trajectories started during the daytime (1-5pm) to avoid computing the residual layer. Hence, we are using much less trajectories now than in the original manuscript. It becomes clear that especially along the edges that are important to the discussion, the number of trajectories is high, hence not many are classified based on just one trajectory being present. We believe this demonstrates that using a threshold of at least one trajectory is a useful approach and we do not substantially misidentify the AD edges. An explaining sentence is added to the manuscript.

**Minor Comments:**

The remaining minor comments that are not addressed here will be addressed directly by encompassing them in the revised manuscript. We will mention the individual changes made in the final response letter.

*L20-21: Is your postulation corroborated by your findings?*
Thank you for raising this concern.
Based on our understanding of the EML literature, we conjecture that ADs should also greatly impact heat wave and thunderstorm formation. Our findings from the case study presented here show that in this case high temperature and thunderstorms did co-occur with the AD event. The results indicated the the AD did influence the formation of the thunderstorms along its edge. However, the high near-surface temperatures were not caused by the lid-processes that was suggested before. Whether they are still caused by the AD or coincidental cannot be definitely answered based on this case study and will be subject to further research.
In order to soften the statement in the manuscript we rephrase it as: "We conjecture that atmospheric deserts (ADs) can greatly impact heat wave and thunderstorm formation."

*L21: Can atmospheric deserts also transport dust towards the mid-latitudes?*
Most certainly, yes. However, if they do will be dependent on the weather situation in the source region. Therefore, dust-bringing ADs, just as EMLs are a subset of ADs. How many of the AD events bring dust will be an interesting question to investigate, once we have a climatology, especially as dusty events may influence the weather in the target region differently than non-dusty events.
We address this in the revised manuscript: "In some cases ADs may also bring dust from the source to the target region, however, this is not analysed in this study."

*L 34-35: The effects of EMLs and ADs are similar. Is there a process in your case study which is new in ADs and not yet found in EMLs?*
We change this to: "The consequences of EMLs and ADs can be expected to be similar, however, the latter was never studied before. Hence this study is looking at one case of an AD that would not have been classified as an EML, but that co-occurred with strong lightning activity along its edge and high near-surface temperatures in its centre." and changed the order of paragraphs in the introduction slightly to accommodate this.
Since this is in the introduction, we do not wish to go into detail of what we found. However, we mainly find that the occurrence of thunderstorms and high near-surface temperatures are indeed similar, but we show that

the mechanism causing the high temperatures in this AD case is not the one that was suggested for EMLs.

*L114: Is there a meteorological reason why you use exactly this region as source region*
North Africa is the source region for typical large scale patterns that bring ADs to Europe. The premise was, therefore, to chose a polygon in Northern Africa as the source region.
Our definition of ADs requires the source region to be hot and dry (with a deep BL). Our entire source region lies within an arid, desert, hot region according to the Koeppen-Geiger climate zone classifications (BWh, see Fig. 1 and Tab. 2 in https://doi.org/10.1038/sdata.2018.214). It can be assumed safely, that if trajectories from further south also play a role, they have to pass through this source region, so we would capture them anyways. We have also investigated the soil types and vegetation cover in the region (in ERA5) and avoided the coastal regions, which have soil types and vegetation indicating that they here the oceanic climate may have a non-negligible influence. See Fig. 4. Additionally, our source region is completely in the Lee of the Atlas mountain range.

*L149: where do the other 80% of the trajectories are going to?*
According to the comment of reviewer 1 we only use trajectories that were started during the daytime, hence the total numbers differ from the original manuscript, the message remains the same, however.
About 45 Mio. trajectories were started during this case study (instead of 200 Mio as stated in the manuscript). 8.7 Mio. of them are used for further analysis (vs 37 before), so you ask about the remaining 80%.
The majority of those never pass 37N and remain over North Africa. The remaining 0.7 Mio. (1.5%) have left the domain by 12UTC on 19 June 2022, so that no information about their location at that time is available and they can therefore not be used in the clustering analysis (independent of whether they have passed over Europe or not). A clarifying sentence is added to the manuscript.

*L164: Which heights are encompassed in the column?*
This is basically a vertical integral, one may also rephrase it and say ERA5-grid cells. I.e. there is at least one trajectory above the respective ERA5-grid cell (or in the column). Effectively, this means up to 13 km, which is the level of the highest detected AD-cell.

*L168: at which height is the majority of ADs?*
At 12:00UTC on 19 June 2022 the majority of the trajectories is at the 500 m layer centred at 2000 m. The majority of detected AD-cells at that time is a the 3500 m level (see Fig. 5 below). This information will also be incorporated in the manuscript.

*L200: Does it make sense to regard Cluster C2 still as an atmospheric desert? Because it leads to precipitation and is not dry anymore?*
This is a question of definition. We do not define ADs by looking for specifically warm and/or dry air masses. The only criterion identifying an AD is its origin being the BL in a semi-arid, desert and/or elevated region. Obviously, one could refine this definition more based on the thermodynamic variables, but we do not think that this will be a helpful approach in understanding the behaviour and consequences of air masses originating in dry and hot boundary layers.

*L203-204: evaporative cooling as precipitation falling … → is this the precipitation from Cluster C2?*
*L207-209: You lost me at this point. C2 is ascending, heated diabatically due to latent heat release, therefore increase in cwc-variables and decrease of q due to precipitation. But why do cwc-variables of C3 act very similar to C2, although C3 is descending and cooled diabatically?*
We believe that the evaporative cooling in C3 is due to precipitation falling through form C2, yes. The similar development of the **precipitation** cwc-variables (cswc, crwc) in the two clusters is an indication for this. When precipitation forms in the upper cluster and falls through the lower one, which did not form its own precipitation, then the precipitation cwc-varibales should naturally be highly correlated. We will make this more clear in the manuscript.

*Section 3.4.1 → this section is a bit disappointing because it does not provide new insights into the formation of heat waves. How long was this heat waves? Typically, a heat wave should at least last three days to be defined as such.*
*L264: the explanation for high surface temperatures of advection and subsidence heating is not really new in the literature … (please review papers on heat wave formation, in particular from a lagrangian point of view)*

Thank you for pointing out your concern. However, the aim of this section is not to give new information about heat wave formation. Several studies about EMLs suggested that EMLs cause near-surface temperatures to rise (see studies cited in the manuscript, and especially Cordeira et al. 2017: https://doi.org/10.1175/WAF-

D-16-0122.1). These studies imply that the EMLs acts as a lid on top of the local BL, which favours clear-sky conditions and prevents the local boundary layer from growing. This can lead to extreme temperatures below. High near-surface temperatures also occur during this case study of an AD. As the AD is also a warm and dry air mass, it could be expected that the reasons for the heat are similar to EML cases. We find, however, that the AD does not reside close to the BLH for long, so it is unlikely that a lid is the reason for the high temperatures in this case. This then raises the question what causes the high temperatures if it is not this.

As we are aware of the literature, we calculate backtrajectories to find out which of the possible processes mentioned in literature may be responsible for the high surface temperatures in this case. Their analysis implies that subsidence heating is a plausible explanation for the high temperatures. We are not suggesting that this is a new insight into the formation of heat waves, as we are aware of the literature you highlighted. We simply find subsidence to be how the high temperatures in this case study can be explained.

There are many ways to classify heat waves based on different measures for strength or length and often one has to chose arbitrary thresholds (like the 3 days you are suggesting). We avoid doing that in this case and just identify anomalously high near-surface temperatures.

*L261-262: perhaps you should look for another period, in which high temperatures persisted at least 3 consecutive days in order to fulfil the criterion of a heat wave? Then you would maybe see that AD form a lid.*
We do not want to prove that in some cases ADs may form a lid, but we wanted to show that it is not the lid that is responsible for high temperatures during this AD event. Therefore, choosing another period will not add anything to the argumentation. In a future climatological study we will address the question how frequently ADs form a lid and what the consequences are.

*L269-L270: Hence, the analysis of the AD event ... → yes, this is correct but it still can be a coincidence, especially when you don't compare this with climatology (either your own or from existing literature on this subject)*
We do not state that it could not be a coincidence.
To make our intent more clear, we rephrase the paragraph in the manuscript as: "Hence, the analysis of the AD event in June 2022 supports the hypothesis that ADs co-occur with anomalously high near-surface temperatures. We find that not the previously suggested mechanism of a lid, but subsidence heating was responsible for the high temperatures in this case. Whether this is caused or facilitated by the presence of the AD, or whether the co-occurrence is coincidental cannot be determined with one case study and will be subject of future research."

*Section 3.4.2: I don't get the message of this section. Is penetration of air into the boundary layer not just a normal process when the boundary layer grows during the day? What is now the special with ADs?*
As the concept of ADs is a generalisation of EMLs, it could be expected that the air is (much) warmer and dryer than the local air and therefore not able to penetrate. Therefore, seeing AD air penetrate was surprising to us, which is why we added this section, explaining why some of the AD air was still able to penetrate the local BL.

*L284: ... while thunderstorm tend to erupt violently along its edges → is this always the case or only under certain circumstances, e.g. a cold front at its edge?*
Literature suggests that thunderstorms tend to erupt at the edges of EMLs. It has been mentioned that in Europe, this may be related to cold fronts in the vicinity of the edge, while in the US it is more often a dry-line (e.g. Carlson and Ludlam, 1968). Pre-frontal convergence lines have been shown to develop in these situations (e.g. Dahl and Fischer, 2016). Whether this is always the case we cannot say. This is why further research in a generalised version of EMLs is necessary with regard to understanding the impact on severe thunderstorms.

*L289: ... the warm AD air still suppresses thunderstorm formation in most parts. → okay, but I assume that the major reason for suppressing the convection is the large-scale subsidence in the anticyclone*
You are right, with the results at hand we cannot disentangle the individual impacts of the AD and the high pressure. We change the sentence in the manuscript to soften the conclusion: "This implies that thunderstorms are suppressed, probably due to a combination of subsidence in the high pressure system and the presence of the (warm) AD aloft."

*Conclusion: a critical assessment of the approach used in this study is lacking* Thank you for pointing this out. In the revised version of the manuscript we will provide a critical discussion.

**Figures**

*Figure 4: Why 00 UTC instead of 12 UTC? 12 UTC perhaps better due to the BLH topography?*
00 UTC was chosen because the lightning activity was higher during this time (see Fig. 3) and this figure is

used to explain the processes behind thunderstorm formation.

Also the remaining comments about the figures will be addressed directly in the revised version of the manuscript and changes will be highlighted in the final response letter.

[Figure]

Figure 1: As Figure 1 in manuscript: AD extent in 800-750 hPa level in shaded contours, maximum extent outlined in black. Fronts and lightning omitted in this case, for better visibility. Colour scale refers to number of trajectories in the respective cell.

[Figure]

Figure 2: As Figure 4 in manuscript: AD air is marked in grey shading in the top 2 panels, the colour scale refers to number of trajectories in the respective grid box.

[Figure]

Figure 3: As Figure 5 in manuscript: AD air is marked in grey shading in the top 2 panels, the colour scale refers to number of trajectories in the respective grid box.

Figure 4: Map showing the source region (red) with ERA5 surface type, surface elevation, high and low vegetation.

[Figure]

Figure 5: Number of trajectories (top, blue) and detected AD-cells (bottom, orange) per 500 m thick layer.

---

## Author Response (AR1)

**Detailed Response Letter**

Fiona Fix and Co-Authors

October 22, 2024

**Answer to the Editor**

Dear Peter Knippertz,

thank you for your immediate response to our previous answer. In response to the reviewers' comments we have improved the manuscript.

The main changes are the following:

- We only use daytime trajectories in the revised version, to avoid using a simplification of the nighttime boundary / residual layer. The results were robust and did not change qualitatively.

- We have clarified that using single trajectories per grid box to identify the as AD-grid boxes is a weak but useful criterion and we do not misidentify many cells, especially at the important edges that are discussed in the study.

- In accordance with your previous comment, we have explained that including a climatology would be too comprehensive for one study. To manage the readers expectations better and emphasise that this study introduces the concept and one case study, we have changed the title and some descriptions in abstract, introduction, and where applicable.

Please find our detailed answers to the reviews below. The reviewers comments are marked in *italic*, and the changes made to the text are highlighted in **bold**.

Kind regards,
Fiona Fix and co-authors

**General Comments**

A new version of Figure 3 is implemented in the manuscript to show the same domain as in Fig 1. A small error in the anomaly calculation was corrected, changes are small and do not influence the discussion.

We updated the code that is available at zenodo by generating a new version, the link provided in the manuscript will always resolve to the most recent version.

**Answer to Reviewer 1**

Dear reviewer,

thank you for your positive feedback and the constructive comments. In the following, we address the comments individually.

**Specific Comments:**

*L58-60: I was wondering how strongly your results depend on the choice that a grid cell is already defined as an AD grid cell when only a single forward trajectory finds its way to the given grid point. I would assume that the fact that the end point of one forward trajectory lies in a grid cell probably does not necessarily mean that the local air mass is then strongly characterized by the AD air mass properties. Maybe this could be tested by initiating a certain number of backward trajectories from an AD grid point? If a majority of these backward trajectories can be traced back to the African source region, then it is safe to say that the local air mass is indeed strongly composed of the AD air.*

Thank you for raising this issue. We had run extensive sensitivity tests at the start of this work by increasing the number of forward trajectories to a couple hundred million. Inspired by your question we have run the experiment again, only using daytime trajectories (see also our answer to your second comment), which drastically reduces the total number of trajectories. Fig. R1 in this review shows that even in this case the vast majority of cells in the layer between 800 and 750 hPa have 10 or more trajectories in them. Similarly, Fig. R4 shows that the determination of the maximum extent of the AD does not hinge on our requirement of only 1 trajectory for the label "AD", since again most columns have 10 or far more trajectories in them. Similarly this can be seen in Figures R2 and R3 included below, which show the vertical cross-sections. Especially in the regions where the location of the AD edge is critical to the discussion, there are many trajectories in the cells, so that a higher minimum number would not change the identification.

Calculating back-trajectories is costly in computing time and data output, and we want to make the detection method feasible to use on long time series of data. Since the explanation above gives trust in the assumption that we do not misclassify many cells by using our approach, we can avoid the extra step of calculating back-trajectories.

We hope, that with this answer we could convince you that labelling a cell as an AD cell when it contains at least one trajectory is a useful approach and likely does not misidentify the AD edges drastically.

**A clarifying sentence has been added to the manuscript: lines 71-75/LatexDiff document: 76-79**

*L68-75: As far as I understand you want to include the nocturnal residual layer air masses for your Lagrangian approach. However, it is in my opinion not fully appropriate to use this kind of a smoothing algorithm because the residual layer might also be of similar height than the daytime convective boundary layer; it might also develop in a certain way over night depending on environmental parameters. The smoothing is unlikely to fully reflect this behavior.*

You are certainly correct, our smoothing approach cannot reflect the true residual layer. The idea was to use a smoothed BLH that still is lower during the night, so we rather under- than overestimate the true residual layer.

*I would suggest to test whether your results would substantially change if you would only use day-time data for which the use of ERA5's boundary layer height might be suitable to estimate the height of the convective boundary layer..*

We recalculated the results using only trajectories that were initiated during the daytime (1-5pm, including 1 and 5pm) and from below the ERA5 BLH. As the results do not substantially change, we have incorporated this in the manuscript now. Note, however, that this reduces the total number of trajectories in total and per cell (see comment before).

**Change to the manuscript:** All the analyses are now shown for only those trajectories started during daytime.

*L80-83: Although I think that I understand how the clustering is performed, it would be beneficial to state a bit more clearly that you are using a multivariate clustering approach. I assume that the data points are clustered within a 11-dimensional space, in which each dimension reflects one of the 11 standardized variables? (such that any of the variables have the same weight)*

You did understand that correctly and **we have added the following sentence to increase clarity:** "We employ the 11 variables listed in Tab. 1 and the data driven k-means-clustering method (MacQueen, 1967) to cluster the trajectories. This is a multivariate clustering approach, using the 11-dimensional data after normalizing all variables (zero mean and unit variance) in order to give all of them the same "weight" or "importance"

in the clustering." (**changes in the revised manuscript: lines 94-99/LatexDiff document: 106-108**)

*L245-248: At least in my opinion, this short paragraph appears to be of minor importance compared to earlier paragraphs. Looking at the plots, I also had problems to identify the air mass that is found between the near-surface air and the overlying AD air mass. Therefore I would suggest to rephrase or remove this short paragraph.*

Indeed, this paragraph is not crucial to the discussion and was therefore **removed**.

*L275-280: I would suggest rewording these three sentences. In their current form, they confused me somewhat because I had difficulty understanding which kind of diabatic processes predominate over other diabatic processes.*

We have **rephrased these sentences to:** "Additionally, the trajectories that did enter the local BL by 19 June 2022, 12:00 UTC, have cooled considerably, while the trajectories that end up above the local BL have warmed (for trajectories initiated on 15 June 2022 the cooling is about 6 K and the warming about 2 K on average). One reason for this different behaviour is that the trajectories entering the local BL experience more radiative cooling along the way on average (not shown here). The development of the specific water contents indicates that another reason for the difference is latent heating. While the trajectories that end up above the local boundary layer seem to form condensate which causes latent heat, the trajectories that enter the local boundary layer cool due to evaporation, likely of precipitation falling through from above (not shown here). "
(**changes in the revised manuscript: lines 303-309/LatexDiff document:330-338**)

**Minor Comments:**

*L19: Parenthesis*
**changed in revised manuscript: line 20/LatexDiff document:22**
*rephrase "thunderstorm eruption"*
**changed where applicable**
*L144 typo*
**changed in revised manuscript: line 164/LatexDiff document:176**
*L313: I think it should say "western edge" instead of "eastern edge" if I am not mistaken*
Thank you for pointing this out to us. Actually, "eastern edge" was correct, but the coordinates were accidentally swapped, which we now corrected. The paragraph before is about the lightning that occurs at the surface front (close to the **western** edge, but this paragraph is about the lightning that occurs close to the **eastern** edge of the AD at the 8800 hPa level. The coordinates were corrected in the text. The content of the discussion was correct, however.
(**changed in revised manuscript: lines 339 and 344/LatexDiff document: 369,375**)
*L346 Diabatic processes*
**changed in the revised manuscript: line 375/LatexDiff Document: 409**

**Answer to Reviewer 2**

Dear reviewer,
thank you for your feedback and the constructive comments. Below, we address your comments individually.

**Major Comments:**

*1. In my view, the title of the study promises a little bit too much. When I first read the title, I imagined that the authors were introducing a new concept and showing how this new concept affects weather in the extratropics in general. In reality, the authors "only" performed a case study, which is not negative in itself, but not enough for this title. This brings me to my main critique of this study. Although the case study is certainly very nicely analysed and interesting, the main findings of the study are not too significant. [...]*

Thank you for this comment. We understand how you might expect more of the study, based on its title. To address your concern, **we adjust the title** to "Detection and Consequences of Atmospheric Deserts: Insights from a Case Study". We **adjust the descriptions** in the abstract, introduction and where appropriate accordingly, in order to not mislead the readers.

Nevertheless, we highlight that the objective of this study is to introduce the new concept of atmospheric deserts based on the idea that they should be a generalisation of EMLs. We base the conceptual interpretation on literature about EMLs and our physical understanding of the involved processes. The case study in this study is used to explain how the direct detection method is applied and to gain first insights into how ADs might be similar or dissimilar to EMLs.

*2. Climatological analysis: I know that this needs very much computing time. But I would suggest that the study would extremely benefit from that. For a climatological analysis a calculation of trajectories on a much coarser resolution would be sufficient (I think this would be feasible in terms of computation time). [...]*

We fully agree that climatological analyses will shed much further light on atmospheric deserts but think that they by far exceed both scope and length of this manuscript. As you suggest, reducing the number of trajectories from our case study will be needed to make a climatological analysis feasible. However, a careful experiment setup is required, as the identification of the AD air mass is sensitive to the amount of initiated trajectories (also see our answer to your major comment 4). We are planning to make use of generalized additive models that have been shown to outperform simply using average cell-counts (i.e. Simon et al. 2017, https://doi.org/10.5194/nhess-17-305-2017). Furthermore, a climatology of AD occurrence "only" is not sufficient to answer all the questions raised in the manuscript. A thorough analysis of the (co-)occurrence of high near-surface temperatures and thunderstorms is required as well. We also plan to use statistical methods to determine whether the presence of an AD is a useful predictor for these extreme events. While this is interesting and necessary research, incorporating all this comprehensive analysis in this paper would result in a very extensive paper, which we do not deem beneficial.

*3. Maybe you can discuss the atmospheric desert with the opposite "atmospheric river"? Both can lead to extreme events, one to extreme precipitation and the other one to extreme temperatures? Maybe you can elaborate on this (also this discussion would need a climatological analysis)*

This is indeed an interesting point. Since a climatological analysis is beyond the scope of this study, we cannot discuss this in much detail. The comparison is not very straight forward, however. While the name "atmospheric deserts" is inspired by "atmospheric rivers", a phenomenon potentially causing severe rainfall (e.g. Ralph et al., 2018), the phenomena are not really each other's opposite. Atmospheric rivers are defined by their water vapour footprint, but atmospheric deserts are defined solely by their source region.
**Added to the discussion in the revised manuscript: lines 398-402/LatexDiff document: 432-435**

*4. I find the definition of an atmospheric desert in L60-61 a bit too weak. Is only one trajectory really enough to significantly affect a grid box? Maybe you could perform a sensitivity analysis and elaborate on that.*

Thank you for this comment, you have a valid point. Reviewer 1 raised a similar concern in their first comment. However, the number of trajectories that reach one cell is not only dependent on whether this cell is dominated by AD air, but also on how many trajectories were initiated. We aim to make this method computationally feasible for longer time series and climatologies, and therefore have to further reduce the amount of trajectories initiated.

We have rerun the experiments with trajectories initiated during the daytime only (in response to reviewer 1), which reduces the total number of trajectory drastically. Even in that case with much less trajectories, Fig. R1 shows that the majority of AD cells in the layer between 800 and 750 hPa have 10 or more trajectories in them. Similarly, most columns have 10 or more trajectories in them, so that the determination of the maximum AD extent does not hinge on the requirement of only one trajectory (see R4). Figs. R2 and R3 show the vertical cross-sections as Figs. 4 and 5 in the initial manuscript. Again, most cells have much more than 10 trajectory

in them. It becomes clear that especially along the edges that are important to the discussion, the number of trajectories is high, hence not many are classified based on just one trajectory being present. We believe this demonstrates that using a threshold of at least one trajectory is a useful approach and we do not substantially misidentify the AD edges. **An explaining sentence is added to the manuscript. lines 72-75/LatexDiff document: 76-79**

**Minor Comments:**

*L20-21: Is your postulation corroborated by your findings?*
Thank you for raising this concern.
Based on our understanding of the EML literature, we conjecture that ADs should also greatly impact heat wave and thunderstorm formation. Our findings from the case study presented here show that in this case high temperature and thunderstorms did co-occur with the AD event. The results indicated the the AD did influence the formation of the thunderstorms along its edge. However, the high near-surface temperatures were not caused by the lid-processes that was suggested before. Whether they are still caused by the AD or coincidental cannot be definitely answered based on this case study and will be subject to further research.
In order to soften the statement **in the manuscript we rephrase it as**: "We conjecture that atmospheric deserts (ADs) can greatly impact heat wave and thunderstorm formation."
(**lines 35-38/LatexDiff Document: 38-40**
Since this sentence is found in the introduction, it serves more as motivation than as conclusion, therefore we do not elaborate further in this paragraph.

*L21: Can atmospheric deserts also transport dust towards the mid-latitudes?*
Most certainly, yes. However, if they do will be dependent on the weather situation in the source region. Therefore, dust-bringing ADs, just as EMLs are a subset of ADs. How many of the AD events bring dust will be an interesting question to investigate, once we have a climatology, especially as dusty events may influence the weather in the target region differently than non-dusty events. The case study presented in this study did indeed bring dust to central Europe.
**We address this in the revised manuscript:** "In some cases ADs may also bring dust from the source to the target region, however, this is not the focus of this study." **lines 50-51/LatexDiff Document: 53-54**
"Aerosol optical depth at 550 nm is acquired to estimate whether the AD transports dust (0 h leadtime for the 00:00 and 12:00 UTC forecast, CAMS; European Centre for Medium-Range Weather Forecasts, 2023)." **lines 146-147 / LatexDiff Doument: 158-159**
"Additionally, in this case the dust aerosol optical depth (550 nm) is notably increased in the area covered by the AD, and especially elevated in its centre. This indicates that the AD brings Sahara dust to the target region, is, however, not further discussed in this paper. " **lines 199-201/ LatexDiffDocument: 214-217**

*L30: which properties?*
**Specified in the manuscript as:** "They occur in the special case where the thermodynamic properties of the AD remain (almost) constant during the advection. " **lines 31/LatexDiff document: 34**

*L 34-35: The effects of EMLs and ADs are similar. Is there a process in your case study which is new in ADs and not yet found in EMLs?*
**We change this** to: "The consequences of EMLs and ADs can be expected to be similar, however, the latter was never studied before. Hence this study is looking at one case of an AD that would not have been classified as an EML, but that co-occurred with strong lightning activity along its edge and high near-surface temperatures in its centre." and changed the order of paragraphs in the introduction slightly to accommodate this. **lines 52-54/ LatexDiff document:55-57**
Since this is in the introduction, we do not wish to go into detail of what we found. However, we mainly find that the occurrence of thunderstorms and high near-surface temperatures are indeed similar, but we show that the mechanism causing the high temperatures in this AD case is not the one that was suggested for EMLs.

*L36-39 For the special case of EMLs ... → add something like "and thereby contributing to potential instability when lifting mechanism is available"*
**Changed to:** "For the special case of EMLs it was found that the hot and dry air masses ride up over the cooler, moister, shallower CBL in the target region, and can form a capping inversion (or "lid", e.g., Carlson and Ludlam, 1968; Carlson, 1980; Carlson et al., 1983; Lanicci and Warner, 1991a, b; Cordeira et al., 2017) and contribute to potential instability. " **lines 42/ LatexDiff document: 45**

*L52-55: this was already mentioned very similarly in the introduction*
Yes, however, this section is giving the definition of atmospheric deserts, which is new in this study, therefore

we think repetition in this case is advantageous.

*Setion 2.1 How long are the forward trajectories and at which pressure/model levels have you initiated them?*
In this section, the principle of the detection method is described. In order to introduce the principle, these details are not necessary and, in fact, rather misleading. This has been made more clear by describing the outline of the paper in the introduction (**changes in revised manuscript: lines 56-62/LatexDiff document: 60-66** ). The exact length of the trajectories and the vertical and horizontal resolution of initiation for the case study presented in the manuscript are given in Section 3.1" "Trajectories are started at a very high resolution of 5 km in the horizontal and 10 hPa in the vertical between 1100 and 400 hPa, from below the BLH between 13:00 and 17:00 UTC. They are calculated 120 h forward in time." **lines 134-136 / LatexDiff: 147**

*Section 3.1: this is better suited in the data and methods section 2.*
Section 2 is not the "data and methods" section, but describes the concept and detection of ADs in general universally applicable terms. In Section 3 this method is then applied, hence it is here, where we introduce the data and exact resolution of the trajectories. We clarified the outline of this manuscript in the introduction (**changes in revised manuscript: lines 56-62/LatexDiff document: 60-66** ).

*L114: Is there a meteorological reason why you use exactly this region as source region*
North Africa is the source region for typical large scale patterns that bring ADs to Europe. The premise was, therefore, to chose a polygon in Northern Africa as the source region.
Our definition of ADs requires the source region to be hot and dry (with a deep BL). Our entire source region lies within an arid, desert, hot region according to the Koeppen-Geiger climate zone classifications (BWh, see Fig. 1 and Tab. 2 in Beck et al., 2018, https://doi.org/10.1038/sdata.2018.214). It can be assumed safely, that if trajectories from further south also play a role, they have to pass through this source region, so we would capture them anyways. We have also investigated the soil types and vegetation cover in the region (in ERA5) and avoided the coastal regions, which have soil types and vegetation indicating that they here the oceanic climate may have a non-negligible influence. See Fig. R5 below. Additionally, our source region is completely in the Lee of the Atlas mountain range.
**Clarified in the manuscript:**
"A polygon marking the source region can be seen as the grey outline in Fig. 1. This source region lies completely in an arid, desert, hot climate zone (Beck et al., 2018), avoids coastal regions, and is in the lee of the Atlas mountains for the flow patterns causing ADs." **lines 130-132/LatexDiff document: 142-144**

*L149: where do the other 80% of the trajectories are going to?*
According to the comment of reviewer 1 we only use trajectories that were started during the daytime, hence the total numbers differ from the original manuscript, the message remains the same, however.
About 45 Mio. trajectories were started during this case study (instead of 200 Mio as stated in the initial manuscript).
8.7 Mio. of them are used for further analysis (vs 37 before), so you ask about the remaining 80%.
The majority of those never pass 37N and remain over North Africa. The remaining 0.7 Mio. (1.5%) have left the domain by 12UTC on 19 June 2022, so that no information about their location at that time is available and they can therefore not be used in the clustering analysis (independent of whether they have passed over Europe or not). **A clarifying sentence is added to the revised manuscript:** "This results in approximately 45 million trajectories starting from the North African BL during this case study. About 80% of the trajectories never pass north of 37° N. About 1.5% have left the domain by 12 UTC on 19 June 2022. The remaining 8.7 million pass north of 37° N at least once and have not left the domain yet by 19 June 2022 12 UTC and are therefore interesting for further analysis." **Lines 168-171/ LatexDiff document:180-183**

*L162: I don't see a warm sector because the low is already occluded.*
You are right, the low pressure system is already occluded. We **changed the sentence in the manuscript to:** "Air from North Africa is advected northwards in the northeasterly current east of this low pressure system." **lines 183/ LatexDiff document: 197**

*L164: Which heights are encompassed in the column?*
This is basically a vertical integral, one may also rephrase it and say ERA5-grid cells. I.e. there is at least one trajectory above the respective ERA5-grid cell (or in the column). Effectively, this means up to 13 km, which is the level of the highest detected AD-cell.
**Clarified in the revised manuscript: lines 185/ LatexDiff document:200**

*L168: at which height is the majority of ADs?*
At 12:00UTC on 19 June 2022 the majority of the trajectories is at the 500 m layer centred at 2000 m. The

majority of detected AD-cells at that time is a the 3500 m level (see Fig. R6 below).
**Incorporated in the revised manuscript lines 190/LatexDiff document: 204-205**

*L186: does not explain this warming → insert "diabatic" warming*
**Inserted in the revised manuscript: Lines 209/ LatexDiff document:225**

*L200: Does it make sense to regard Cluster C2 still as an atmospheric desert? Because it leads to precipitation and is not dry anymore?*
This is a question of definition. We do not define ADs by looking for specifically warm and/or dry air masses. The only criterion identifying an AD is its origin being the BL in a semi-arid, desert and/or elevated region. Obviously, one could refine this definition more based on the thermodynamic variables, but we do not think that this will be a helpful approach in understanding the behaviour and consequences of air masses originating in dry and hot boundary layers.

*L203-204: evaporative cooling as precipitation falling ... → is this the precipitation from Cluster C2?*
*L207-209: You lost me at this point. C2 is ascending, heated diabatically due to latent heat release, therefore increase in cwc-variables and decrease of q due to precipitation. But why do cwc-variables of C3 act very similar to C2, although C3 is descending and cooled diabatically?*
We believe that the evaporative cooling in C3 is due to precipitation falling through from C2, yes. The similar development of the **precipitation** cwc-variables (cswc, crwc) in the two clusters is an indication for this. When precipitation forms in the upper cluster and falls through the lower one, which did not form its own precipitation, then the precipitation cwc-varibales should naturally be highly correlated.
**We make this more clear in the manuscript:** "In contrast, cluster C3 (red) remains at a constant height above mean sea level after the ascent during the initial 24 h and then experiences a descent around hour 80. Meanwhile, its potential temperature decreases (Fig. 2b) and its specific water vapour content increases (Fig. 2d). This is partly due to radiative cooling (dashed in Fig. 2b), and partly due to evaporative cooling as precipitation falling through from above re-evaporates. This explanation is supported by the fact that together with the decrease in potential temperature (Fig. 2b), the specific water vapour content increases (Fig. 2d). Re-evaporation is possible in the data used here, since ice, snow and rain are allowed to sediment and can re-evaporate when they fall through a sub-saturated air mass in ERA5 (European Centre for Medium-Range Weather Forecasts, 2016). The strong correlation between the precipitation cloud water contents of C2 and C3 (dashed in Fig. 2e, f) together with the increase of specific water content (Fig. 2d) indicates it may be the precipitation from C2 that re-evaporates in C3." **lines 225 ff /LatexDiff document: 241 ff**

*L245: above the cold air mass at the surface → do you mean at around 12°W?*
*L247: repetitive to L241 and following lines*
**In response to reviewer 1 this paragraph is cut from the revised manuscript, as is it not crucial to the discussion.**

*Section 3.4.1 → this section is a bit disappointing because it does not provide new insights into the formation of heat waves. How long was this heat waves? Typically, a heat wave should at least last three days to be defined as such.*
*L264: the explanation for high surface temperatures of advection and subsidence heating is not really new in the literature ... (please review papers on heat wave formation, in particular from a lagrangian point of view)*
Thank you for pointing out your concern. However, the aim of this section is not to give new information about heat wave formation. Several studies about EMLs suggested that EMLs cause near-surface temperatures to rise (see studies cited in the manuscript, and especially Cordeira et al. 2017: https://doi.org/10.1175/WAF-D-16-0122.1). These studies imply that the EML acts as a lid on top of the local BL, which favours clear-sky conditions and prevents the local boundary layer from growing. This can lead to extreme temperatures below. High near-surface temperatures also occur during this case study of an AD. As the AD is also a warm and dry air mass, it could be expected that the reasons for the heat are similar to EML cases. We find, however, that the AD does not reside close to the BLH for long, so it is unlikely that a lid is the reason for the high temperatures in this case. This then raises the question what causes the high temperatures if it is not this.
As we are aware of the literature, we calculate backtrajectories to find out which of the possible processes mentioned in literature may be responsible for the high surface temperatures in this case. Their analysis implies that subsidence heating is a plausible explanation for the high temperatures. We are not suggesting that this is a new insight into the formation of heat waves, as we are aware of the literature you highlighted. We simply find subsidence to be how the high temperatures in this case study can be explained.
There are many ways to classify heat waves based on different measures for strength or length and often one has to chose arbitrary thresholds (like the 3 days you are suggesting). We avoid doing that in this case and just identify anomalously high near-surface temperatures, without quantifying the heat wave.

**Section 3.4.1 was refined to make the intent more clear**

*L261-262: perhaps you should look for another period, in which high temperatures persisted at least 3 consecutive days in order to fulfil the criterion of a heat wave? Then you would maybe see that AD form a lid.*
We do not want to prove that in some cases ADs may form a lid, but we wanted to show that it is not the lid that is responsible for high temperatures during this AD event. Therefore, choosing another period will not add anything to the argumentation. In a future climatological study we will address the question how frequently ADs form a lid and what the consequences are.

*L269-L270: Hence, the analysis of the AD event ... → yes, this is correct but it still can be a coincidence, especially when you don't compare this with climatology (either your own or from existing literature on this subject)*
We do not state that it could not be a coincidence.
To make our intent more clear, we **rephrase the paragraph in the manuscript as:** "Hence, the analysis of the AD event in June 2022 supports the hypothesis that ADs co-occur with anomalously high near-surface temperatures. We find that not the previously suggested mechanism of a lid, but subsidence heating was responsible for the high temperatures in this case. Whether this is caused or facilitated by the presence of the AD, or whether the co-occurrence is coincidental cannot be determined with one case study and will be subject of future research." **lines 293-297/LatexDiff document: 320-324**

*Section 3.4.2: I don't get the message of this section. Is penetration of air into the boundary layer not just a normal process when the boundary layer grows during the day? What is now the special with ADs?*
As the concept of ADs is a generalisation of EMLs, it could be expected that the air is (much) warmer and dryer than the local air and therefore not able to penetrate. Therefore, seeing AD air penetrate was surprising to us, which is why we added this section, explaining why some of the AD air was still able to penetrate the local BL. We clarify this in the manuscript: "In the centre of the AD air mass, AD air penetrates the local BL, which may be surprising, considering that the AD air can be expected to be warmer and dryer than the local BL air." **line 209 / LatexDiffDocument: 326**

*L284: ... while thunderstorm tend to erupt violently along its edges → is this always the case or only under certain circumstances, e.g. a cold front at its edge?*
Literature suggests that thunderstorms tend to erupt at the edges of EMLs. It has been mentioned that in Europe, this may be related to cold fronts in the vicinity of the edge, while in the US it is more often a dry-line (e.g. Carlson and Ludlam, 1968). Pre-frontal convergence lines have been shown to develop in these situations (e.g. Dahl and Fischer, 2016). Whether this is always the case we cannot say. This is why further research in a generalised version of EMLs is necessary with regard to understanding the impact on severe thunderstorms. **added to the manuscript:** "It has been described for EMLs that the capping inversion due to the warm, well-mixed layer aloft suppresses thunderstorm formation in the centre of the EML, while thunderstorms tend to break out violently along its edges (often in the vicinity of cold fronts or dry-lines; Carlson and Ludlam, 1968; Carlson et al., 1983; Farrell and Carlson, 1989; Keyser and Carlson, 1984; Dahl and Fischer, 2016). " **lines 314 / LatexDiff Document 344**

*L289: ... the warm AD air still suppresses thunderstorm formation in most parts. → okay, but I assume that the major reason for suppressing the convection is the large-scale subsidence in the anticyclone*
You are right, with the results at hand we cannot disentangle the individual impacts of the AD and the high pressure. **We change the sentence in the manuscript to soften the conclusion:** "This implies that thunderstorms are suppressed, probably due to a combination of subsidence in the high pressure system and the presence of the (warm) AD aloft." **lines 318-320/ LatexDiff document:348-349**

*L310: ... while the equivalent potential temperature decreases with height → very hard to see in the figure*
For better visibility we display the $\theta_E$ gradient in shading now in panel (b) of **Fig 4.**

*Conclusion: a critical assessment of the approach used in this study is lacking* Thank you for pointing this out. **A critical discussion is added to the manuscript. Lines 403ff / in Latex Diff document:436**

We did review the suggested literature on formation of heat waves. They are also cited now in Section 3.4.1.

**Figures**

*Figure 1: Display of the situation during the second half ... → what is the first half, since you analyse 15-19 June 2022?*

*Figure 1: 800 hPa fronts → how were fronts identified?*
**Changed to:** "Figure 1. Display of the situation during 16–19 June 2022, 00:00 UTC, respectively. Thin black contours show the 800 hPa geopotential height in decametres, with a spacing of 3 dam. The coloured lines denote the 800 hPa fronts (identified from 800 hPa temperature, relative humidity, divergence, and relative vorticity maps), colours and symbols have their usual meaning. The maximum extent of the AD is outlined in thick black. The extent of the AD in the layer from 800 to 750 hPa is marked in beige. Yellow crosses mark locations where lightning occurred during the hour before and after. Red line (A–B) marks the locations of the cross-section depicted in Fig. 4."

*Figure 2: are the initial values at (b), (c), (d) relevant? Because you don't inserted the initial values at (a), (e) and (f).*
The initial values were printed in the panels that showed relative rather than absolute values on the y-axis, to not lose this information. For completeness we **add the initial values in panel (a)** as well. In Panels (e) and (f) this would however result in a very cluttery plot and since it is not necessary for the discussion we decided to leave these out.

*Figure 3: 00 UTC instead of 06 UTC, What do the red lines show?*
**Changed to**: "Figure 3. Map showing the spatial extent of the AD and the 2 m temperature anomaly with respect to the 30 year period 1992–2021 at 00:00 UTC and 12:00 UTC on 18 and 19 June 2022. The entire AD is outlined in black, outlines of the AD cells up to 4 and 2 km, respectively, are marked in grey. White crosses mark locations where lightning occurred during the hour before and after, 2 m temperature anomalies are coloured with 2 K spacing. Red line (A–B) marks the locations of the cross-section depicted in Fig. 4, as in Fig. 1."

*Figure 4: Why 00 UTC instead of 12 UTC? 12 UTC perhaps better due to the BLH topography?*
00 UTC was chosen because the lightning activity was higher during this time (see Fig. 3) and this figure is used to explain the processes behind thunderstorm formation.

*Figure 5 doesn't provide many new insights → perhaps a cross-section in the area from Fig. 6 would be better to further the insights on the penetration of AD air into the local boundary layer?*
In response to your comment we **removed Fig. 5**, which does not provide many new insights. We have **adjusted the paragraph**:
"A longitudinal cross-section 3° E, 42° N to 3° E, 59° N, helps understanding the lightning that occurs between 51.5° and 53.25° N (not shown here). This region experiences strong upward motion above 2 km, and is located north of the surface front, but south of the 800 hPa front and the AD edge at that level. Also here there is a zone of convergence (not shown here). Centred at the region with strongest upward motion, there is a wider region where ERA5 produces both large scale and convective precipitation in response to the lifting. In this case, there is increased CAPE at the locations of the thunderstorms, which is released. However, the rising motion is not surface-based, which it should be if underrunning were the reason for it." **lines 353ff / LatexDiff: 384ff**
Figures R7 and R8 below show the same cross-sections as Fig 4. in the manuscript, but at 51 and 44 degrees north, and 12 UTC on June 19 and 18, respectively. They therefore show cross-sections through the regions from Fig 6. (Fig. 5 in revised version) in the manuscript. It becomes obvious that the local BL is very deep in these regions and that the AD penetrated into the local BL. However, this does not give further insight into why the BL is so warm and how the trajectories enter.

*Figure 6: Back-trajectories started at which level?*
**The following sentence was added in the text (lines 285-290 /LatexDiff document: 312-316):** (Fig 6 is now Fig. 5)
"To investigate the involved processes in this case more closely, back-trajectories from the BL in two regions in eastern Germany and southwestern France which experienced exceptionally high temperatures (Imbery et al., 2022) were calculated (see Fig. 5). As for the forward trajectories, the trajectories are initiated from the defined region, at a horizontal resolution of 5 km in the horizontal and 10 hPa in the vertical between 1100 and 400 hPa. Only those that are initially below the local BLH are then used for further analysis."

**Additional Figures to Support the Answers to the Reviewers**

[Figure]

Fig. R1: As Figure 1 in manuscript, but only for daytime trajectories and with numbers of trajectories per cell in colour: AD extent in 800-750 hPa level in shaded contours , maximum extent outlined in black. Fronts and lightning omitted in this case, for better visibility. Colour scale refers to number of trajectories in the respective cell (#trajectories per cell of 0.25°x0.25°x50hPa at the 800-750hPa level).

[Figure]

Fig. R2: As Figure 4 in manuscript, but only for AD air is marked in grey shading in the top 2 panels, the colour scale refers to number of trajectories in the respective grid box.

[Figure]

Fig. R3: As Figure 5 in initial manuscript, but AD air is marked in grey shading in the top 2 panels, the colour scale refers to number of trajectories in the respective grid box.

[Figure]

Fig. R4: Colour shading for number of trajectories in the respective column, i.e. showing the maximal AD extent (as marked in black outline in R1). Attention: extended colour scale compared to Fig. R1.

[Figure]

Fig. R5: Map showing the source region (red) with ERA5 surface type, surface elevation, high and low vegetation.

[Figure]

Fig. R6: Number of trajectories (top, blue) and detected AD-cells (bottom, orange) per 500 m thick layer.

[Figure]

Fig. R7: As Fig 4 in the manuscript, but at 51 degrees north, 12 UTC on June 19. Green vertical lines mark the region from which backtrajectories are initiated, see Fig. 5a (formerly 6a) in the manuscript.

[Figure]

Fig. R8: As Fig 4 in the manuscript, but at 44 degrees north, 12 UTC on June 18. Green vertical lines mark the region from which backtrajectories are initiated, see Fig. 5b (formerly 6b) in the manuscript.